# Coupling of growth rate and developmental tempo reduces body size heterogeneity in *C. elegans*

Klement Stojanovski[1], Helge Großhans [2,3✉] & Benjamin D. Towbin [1,2✉]

Animals increase by orders of magnitude in volume during development. Therefore, small variations in growth rates among individuals could amplify to a large heterogeneity in size. By live imaging of *C. elegans*, we show that amplification of size heterogeneity is prevented by an inverse coupling of the volume growth rate to the duration of larval stages and does not involve strict size thresholds for larval moulting. We perturb this coupling by changing the developmental tempo through manipulation of a transcriptional oscillator that controls the duration of larval development. As predicted by a mathematical model, this perturbation alters the body volume. Model analysis shows that an inverse relation between the period length and the growth rate is an intrinsic property of genetic oscillators and can occur independently of additional complex regulation. This property of genetic oscillators suggests a parsimonious mechanism that counteracts the amplification of size differences among individuals during development.

[1] University of Bern, Bern, Switzerland. [2] Friedrich Miescher Institute for Biomedical Research (FMI), Basel, Switzerland. [3] University of Basel, Basel, Switzerland. ✉email: helge.grosshans@fmi.ch; benjamin.towbin@unibe.ch

Given the large volume increase that animals undergo during development, even small differences in the growth rate among individuals could, in principle, amplify to large differences in their final size (Fig. 1a). However, individuals of the same species differ only little in their adult body volume, and growth to the appropriate size is indeed under strong selective pressure[1]. Although molecular pathways that promote or limit the growth of cells, organs, or organisms have been characterized extensively[2,3], the mechanisms responsible for size uniformity among individuals are poorly understood.

Studies of animal size control in the context of environmental and genetic perturbations led to the hypothesis that specific size thresholds are associated with the passing of developmental milestones[2,3]. For example, insects commit to metamorphosis at a critical weight[4]. Similarly, the body volume of *C. elegans* at larval stage transitions is nearly invariant under dietary restriction[5], and growth retardation of humans due to malnutrition or hormonal

**Fig. 1 *C. elegans* grows faster than linearly within larval stages. a** Illustration of volume divergence during exponential growth. The blue individual has a 10% faster growth rate than the red individual (0.11/h vs. 0.10/h). **b** Images of the same individual at birth and moult 1 (M1) to moult 4 (M4). Dotted square indicates edge of the chamber with dimensions of 600 μm × 600 μm. The red line shows the segmentation outline. Images at the bottom are computationally straightened animals used for volume computation shown in Fig. 1e. **c** Volume measurement of one individual as a function of time starting at hatching. Arrows indicate larval stage transitions detected as restart of growth after lethargus. **d** Average absolute ($\mu_{abs}$) and relative ($\mu_{rel}$) growth rate during development. The x-axis indicates the fraction $\tau$ of larval stage progression ($\tau$ = [time since last moult]/[total duration of larval stage]). Average growth rates were computed by re-scaling individuals according to moulting times and averaging growth rates across all individuals at the same $\tau$. Dips in the growth rate correspond to growth halt during lethargus. Dashed lines are 95% confidence intervals for the mean of day-to-day repeats. **e** Box plots of larval stage duration, volume, and volume fold change in micro chambers. Grey dots are individuals, blue lines: median, red dots: mean, box: interquartile ranges (IQR), whisker: ranges except extreme outliers (>1.5*IQR), $n$ = 639, 1142, 1144, 1095 (for L1 to L4) biologically independent animals examined over 10 independent experiments.

imbalance is compensated by catch-up growth later in life[6,7]. However, measuring the effects of exogenous growth perturbation on body volume does not necessarily inform on the mechanisms of size uniformity among individuals. Understanding what drives and counteracts the differences among individuals requires precise measurement of individual animals at high throughput[8–11], which has remained challenging at a multicellular scale.

For unicellular systems, the correlative analysis of cell size at birth and division has distinguished so-called sizer and adder mechanisms of cell size homeostasis[12–16]. A sizer is defined as a mechanism involving a size threshold for cell division, such that the size at cell division is independent of the size at cell birth[17–19]. Cells following an adder mechanism grow by a constant absolute volume in each cell cycle[12–16,20–22]. Sizers as well as adders have been observed for bacteria, yeasts, and mammalian cells. Which of these mechanisms dominates, depends on the cell type, as well as on environmental conditions[12–16,20–22]. Importantly, both mechanisms converge to a stable cell size over one or several generations. Whether adders and sizers also function at an organismal scale in multicellular animals was not known prior to this study.

The adult body volume of multicellular organisms is determined by multiple parameters, including the growth rate (i.e., the rate of volume increase), the duration of development (i.e., the time during which growth occurs before adulthood is reached), and the volume at birth. In principle, fluctuations in these parameters can be correlated, anti-correlated, or be independent of each other. As such, fluctuations in two parameters can either cancel each other out or add up to amplify body volume differences. For example, a slowly growing individual can reach the same volume as a rapidly growing individual if its development is sufficiently slowed down, providing more time for body growth to occur.

In this study, we used C. elegans to measure the heterogeneity of volume growth among individuals. C. elegans is well-suited for this purpose, given its stereotypic and precisely characterized development through four larval stages (L1 to L4). The duration of C. elegans larval stages is thought to be controlled by a developmental clock with oscillatory transcriptional output[23–25]. Although the molecular mechanism driving these oscillations remains incompletely characterized, the output of the developmental clock has been well described: the clock controls the oscillatory expression of >3000 genes. All genes share the same oscillatory period of roughly 8 h[24]. From L2 to L4, oscillations are in synchrony with larval stage duration, i.e., each larval stage takes as long as the period of one oscillation. The L1 stage takes longer than one oscillatory period, since the oscillator is arrested during the first hours after hatching[23]. Notably, different genes peak at different times, depending on when they function during the moulting cycle of a larval stage. For example, the phase-shifted oscillatory expression of different cuticular collagens ensures their timely synthesis prior to moulting[24]. Hence, development and oscillations are tightly connected[23]: conditions that change the duration of development also change the frequency of oscillations, and vice versa[24].

Using live imaging of hundreds of individuals, we characterized the sources of body volume heterogeneity. We found that, unlike unicellular systems, the total body volume of C. elegans follows neither a sizer nor an adder mechanism. Instead, the volume fold change within one larval stage was nearly invariant with respect to the volume at the larval stage entry. Despite the lack of strong size-dependent growth control by adders or sizers, we observe very little divergence of body volume between rapidly and slowly growing individuals. This uniformity of body size is explained by an anti-correlation of the growth rate and the duration of development. Using a mathematical model of developmental oscillations, we show that an inherent dependence of the oscillation frequency on the growth rate can explain this coupling between growth and development. Consistent with model predictions, experimental alteration of the oscillation frequency uncoupled developmental duration from the growth rate and changed body volume.

## Results

**Measurement of growth and body size of individual C. elegans larvae**. To measure the growth rate and the body size of C. elegans, we used agarose-based microchambers[23,26] to track individual animals at high temporal resolution throughout post-embryonic development by live imaging. We used a strain ubiquitously expressing *gfp* under control of the *eft-3* promoter from a single-copy transgene[23], providing high contrast images for robust and precise segmentation of the outline of animals (Fig. 1b). We placed individual embryos of this strain in arrayed chambers filled with the bacterial strain E. coli OP50, the standard laboratory diet of C. elegans, and sealed the chambers by adherence to a gas permeable polymer. This experimental setup allowed us to image up to 250 individuals of C. elegans in parallel at a time resolution of 10 min from hatch to adulthood by fluorescence microscopy. The temperature was kept constant at 25 °C ± 0.1 °C by enclosing the microscope in a dedicated incubator. In total, we collected data of 1,153 individuals from 12 micro chamber arrays that were imaged on 10 different days.

We determined the body volume at each time point from 2D images, by computationally straightening the central focal plane of each worm after segmentation and assuming rotational symmetry, as previously described[27] (Fig. 1b). Consistent with previous observations[5], we observed four plateaus with near absent or negative volume growth for ~2 h, followed by a saltatory increase in body volume (Fig. 1c, d). This halt in growth was accompanied by a lethargic period prior to cuticular ecdysis[28,29], during which animals stopped feeding with a constricted pharynx and intestine[28] (Supplementary Fig. 1a). We could thereby determine larval stage transitions as the timepoints of growth restart after each lethargic episode. The average larval stage durations determined using this assay were close to manual assignments in microchambers[23], and on standard petri-dishes[30] (mean duration was 13.1 h, 8.0 h, 7.7 h, 10.4 h for L1-L4). On average, animals increased 40-fold in volume between hatch and the fourth moult, with 2- to 3-fold volume changes per larval stage. Volume fold changes were larger in L1 and L4 stages than in L2 and L3, consistent with the longer duration of these two stages (Fig. 1e).

***C. elegans* grows faster than linearly within larval stages**. Body volume uniformity among individuals is especially sensitive to changes in the growth rate during exponential growth, where the absolute volume increase per time is proportional to the current volume. Hence, under exponential growth, the difference in volume between two individuals with different growth rates amplifies exponentially with time (Fig. 1a). Previous research has described growth of C. elegans as piecewise linear, with constant linear growth within a larval stage and a saltatory increase in the linear growth rate upon larval stage transitions[5,31]. However, distinguishing between linear and exponential growth within a larval stage requires highly accurate measurements[32] and measurements at high temporal resolution are needed to avoid confounding effects of growth pauses during lethargus.

To re-evaluate if the body volume of C. elegans increases linearly, or faster than linearly within a larval stage, we determined the average absolute rate of volume increase ($\mu_{abs}(t) = \frac{dV(t)}{dt}$.) and the rate of volume increase normalized to

the current volume $(\mu_{rel}(t) = \frac{dV(t)}{dt} * \frac{1}{V(t)} = \frac{d\ln(V(t))}{dt})$ within each larval stage. To this end, we divided the larval stages of each individual into 100 equally spaced intervals and averaged $\mu_{abs}$ and $\mu_{rel}$ in each interval over all individuals. This analysis yielded the growth rate as a function of larval stage progression $\tau$. Consistent with supra-linear growth, the mean absolute growth rate $<\mu_{abs}(\tau)>$ increased monotonically during development with the exception of the lethargic periods immediately prior to ecdysis (Fig. 1d). The average relative growth rate $<\mu_{rel}(\tau)>$ was nearly constant within larval stages L1 to L3, consistent with exponential growth. Growth during the L4 stage was also faster than linear (Fig. 1d), but slower than exponential, as $\mu_{rel}$ declined towards the end of the larval stage (Fig. 1d). Notably, this decrease of $\mu_{rel}$ coincided with a previously reported slowing down of the oscillatory developmental clock at the end of L4[23]. The decline of relative growth rate during L4 is therefore likely a developmental feature of this larval stage, although we cannot entirely exclude an influence of geometric or other constraints on $\mu_{rel}$ during the L4 stage.

To exclude that supra-linear growth within a larval stage was caused by re-scaling and averaging, we fitted a linear and an exponential growth model to each individual. We excluded the first 10% and the last 25% of each larval stage to avoid confounding effects of the growth halt during lethargus and the saltatory volume increase after moulting. For L1 to L3 stages, more than 90% of the individuals had a better fit to the exponential growth model (higher pearson correlation coefficient between fit and data for 98%, 99%, and 93% of individuals of L1-L3, $p < 10^{-50}$ bionomial test). For the L4 stage, linear growth provided a better fit than exponential growth for the majority of individuals (83%, $p < 10^{-50}$, binomial test), confirming that, although growth is faster than linear during the L4 stage (Fig. 1d), the growth dynamics during L4 cannot be captured by a linear nor by an exponential growth model.

Finally, for all larval stages, individuals with a large body volume at the beginning of the larval stage had a faster absolute rate of volume increase $\mu_{abs}$ than smaller animals of the same larval stage (Supplementary Fig. 1b, c), as is expected for autocatalytic growth. We conclude that $C. elegans$ grows faster than linearly during all larval stages, raising the challenge of amplifying heterogeneity in body volume during development due to differences in the growth rate. In the following, we will refer to $\mu_{rel}$ as the *growth rate* without specifying normalization to the current volume at every use. $\mu_{abs}$ will be specified as the *absolute growth rate*.

**Maintenance of body size uniformity despite growth rate heterogeneity.** We next asked how much individuals of $C. elegans$ differed among each other in their growth rate. Heterogeneities among individuals can either stem from batch effects of day-to-day repeats, or from heterogeneity among individuals from the same batch. Despite highly standardized experimental conditions, we observed small but significant differences between batches that may stem from microenvironmental, parental[8], or other effects. To exclude any influence of batch effects on our conclusions, we normalized growth and size relative to the mean of the population of each respective day. For the rest of this article, main figures show results from batch corrected data, whereas the related Supplementary Figures will show that the trends hold even without batch correction, in the unnormalized data.

After normalization, the coefficient of variation (CV) in the growth rate ranged between 6% and 12%, depending on the larval stage (Fig. 2a, b). For a given individual, the differences in the growth rates persisted between consecutive larval stages (Supplementary Fig. 2a). Autocorrelation was reduced when comparing

larval stages further apart and was nearly absent when comparing L1 with L4 larvae. Hence, under the conditions studied here, genetically identical individuals of $C. elegans$ display heterogeneity in their growth rate that is partially transmitted between larval stages with decay of the autocorrelation on a time scale of days. We observed a similar correlation for the duration of the larval stages across development (Supplementary Fig. 2b), consistent with previous observations[30]. Positive autocorrelation of growth rates also indicates that $C. elegans$ does not undergo catch-up growth, where slow growth early in development would be compensated by fast growth later in development[7]. However, we cannot entirely exclude a contribution of micro environmental effects on the autocorrelation among individuals[33].

To ask if differences in the growth rate accumulate to differences in volume during development, we next determined the heterogeneity in body volume at each larval stage. At birth, the CV of the body volume was 6.1% and increased moderately to 9.6% at L4-to-adult transition (Fig. 2c, d and Supplementary Figs. 2d, e). This increase in body volume heterogeneity was significantly smaller than expected based on random shuffling of growth rates and larval stage durations (19.1% at L4-to-adult transition, $p < 10^{-5}$, ranksum test, Fig. 2c, d and Supplementary Figs. 2d, e, see methods). Volume divergence among individuals at 20 °C was even smaller than at 25 °C, consistent with 25 °C being close to the upper bound of the temperature range compatible with robust control of body size of $C. elegans$. (Supplementary Fig. 3). Together, our measurements imply a mechanism that buffers body volume against heterogeneities in the growth rate.

**$C. elegans$ does not follow a sizer or an adder, but a folder mechanism.** A frequently proposed mechanism for size uniformity involves size thresholds that gate the passing of developmental milestones[4–7]. For bacteria, yeasts, and individual mammalian cells, analysis of correlations between size at birth ($V_1$) and at division ($V_2$) of individual cells have indeed suggested a range of mechanisms involving size thresholds[12–17,19–21] (Fig. 3). Specifically, a sizer is defined as a mechanism that triggers cell division at a fixed volume threshold. Since this threshold is independent of the starting volume $V_1$, $V_2$ and $V_1$ are uncorrelated. An *adder* is defined as a mechanism where cell division occurs after adding a fixed absolute volume increase ($\Delta V = V_2 - V_1$), such that $\Delta V$ is uncorrelated with $V_1$. Like sizers, adders converge to a stable size distribution, albeit only over multiple cell division[12–17,19–21].

Using single-animal measurements, we asked if the larval stages of $C. elegans$ also follow adder or sizer mechanisms. As described above, we normalized volumes for each individual and stage to the respective population mean separately for each day-to-day repeat to minimize batch effects, and to facilitate comparisons between larval stages with different volumes. Importantly, the normalization does not impact the expected lack of correlation between $V_1$ and $V_2$ for sizers, or between $V_1$ and $\Delta V$ for adders (Supplementary Fig. 4).

For $C. elegans$, we find no evidence for a sizer mechanism, since $V_1$ (the volume at larval stage entry) was positively correlated with $V_2$ (volume at larval stage exit) for all larval stages (Fig. 4a). The L1 stage was close to an adder, with $\Delta V$ being nearly independent of $V_1$ (Fig. 4b). However, in L2 to L4 stages, we observed a different mechanism. Here, $V_2$ as well as $\Delta V$ were positively correlated with $V_1$ (Fig. 4b, Supplementary Fig. 5), and we observed near independence between $V_1$ and the volume fold change $FC_V = V_2/V_1$, except for the smallest L2 larvae (Fig. 4c, Supplementary Fig. 5). These trends were robust to changes in temperature: consistent with our measurements at 25 °C, $V_2$ and

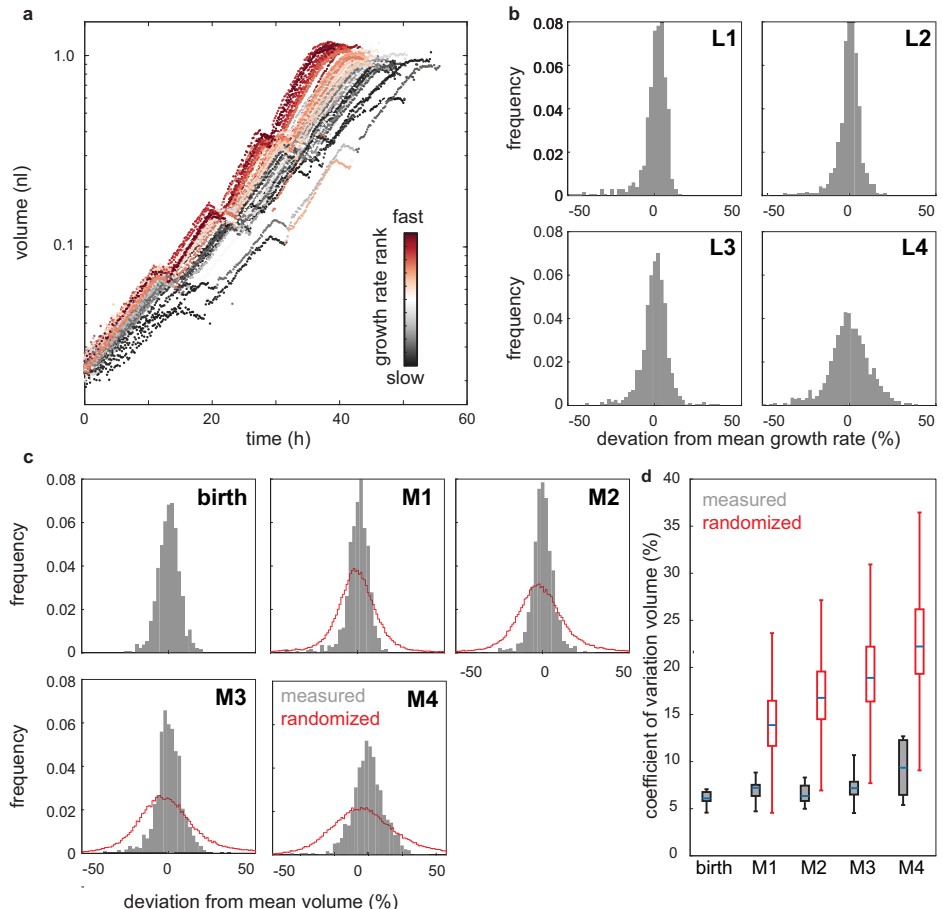

**Fig. 2 Volume divergence among individuals is less than expected by the heterogeneity in growth rates. a** Volume trajectories for all individuals measured in one experimental repeat (*n* = 86). Colours represent the rank of growth rate at each larval stage (grey = slowest, red = fastest). **b** Histogram of the growth rate deviations in % from the population mean measured at indicated larval stages. The growth rate of each individual and larval stage was determined by a linear regression to ln(volume) against time excluding the first 10% and the last 25% of the larval stage. **c** Histogram of body volume deviations in % from the mean measured at birth and indicated larval moults. Red line shows distribution of volume resulting from random shuffling of growth rate and larval stage duration among individuals. **d** Coefficient of variation of volume during development. grey: box plot of the average CV measured experimentally in the different day-to-day repeats. red: expected CV based on 1000x random shuffling of growth rate and larval stage duration. The measured CV was significantly smaller than the randomized control at all stages ($p = 2*10^{-7}$, $6*10^{-8}$, $6*10^{-8}$, $8*10^{-8}$, two-sided ranksum test for M1–M4). $n = 10$ biologically independent experiments. Boxplots: central line: median, box: interquartile ranges (IQR), whisker: ranges except extreme outliers (>1.5*IQR).

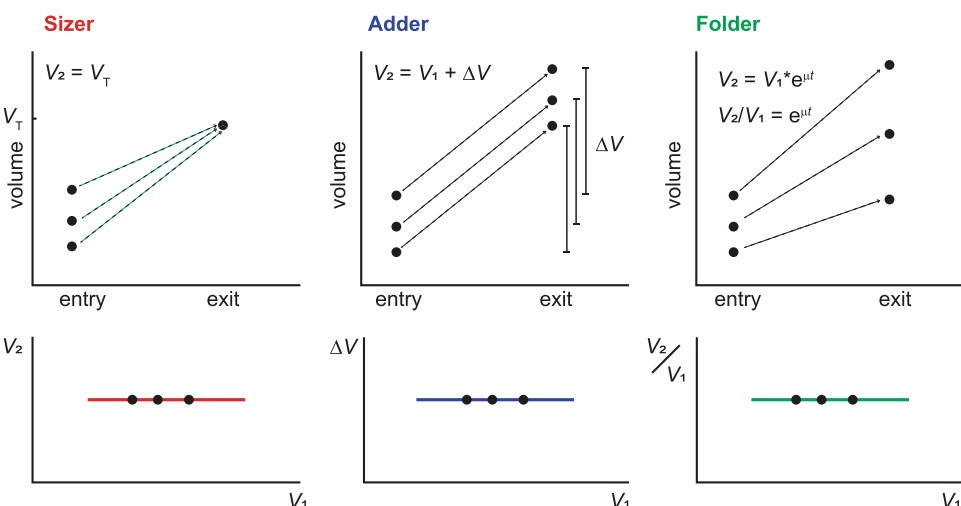

**Fig. 3 Definition of sizer, adder, and, folder mechanisms.** A sizer is defined as a mechanism where the volume at the larval stage exit is independent of the volume at larval stage entry. For an adder, the absolute added volume within a larval stage is independent of the volume at the beginning of the larval stage. For a folder, the volume fold change per larval stage is independent of the volume at larval stage entry. A folder is expected during exponential growth if the exponential growth rate and duration of growth are independent of the current size.

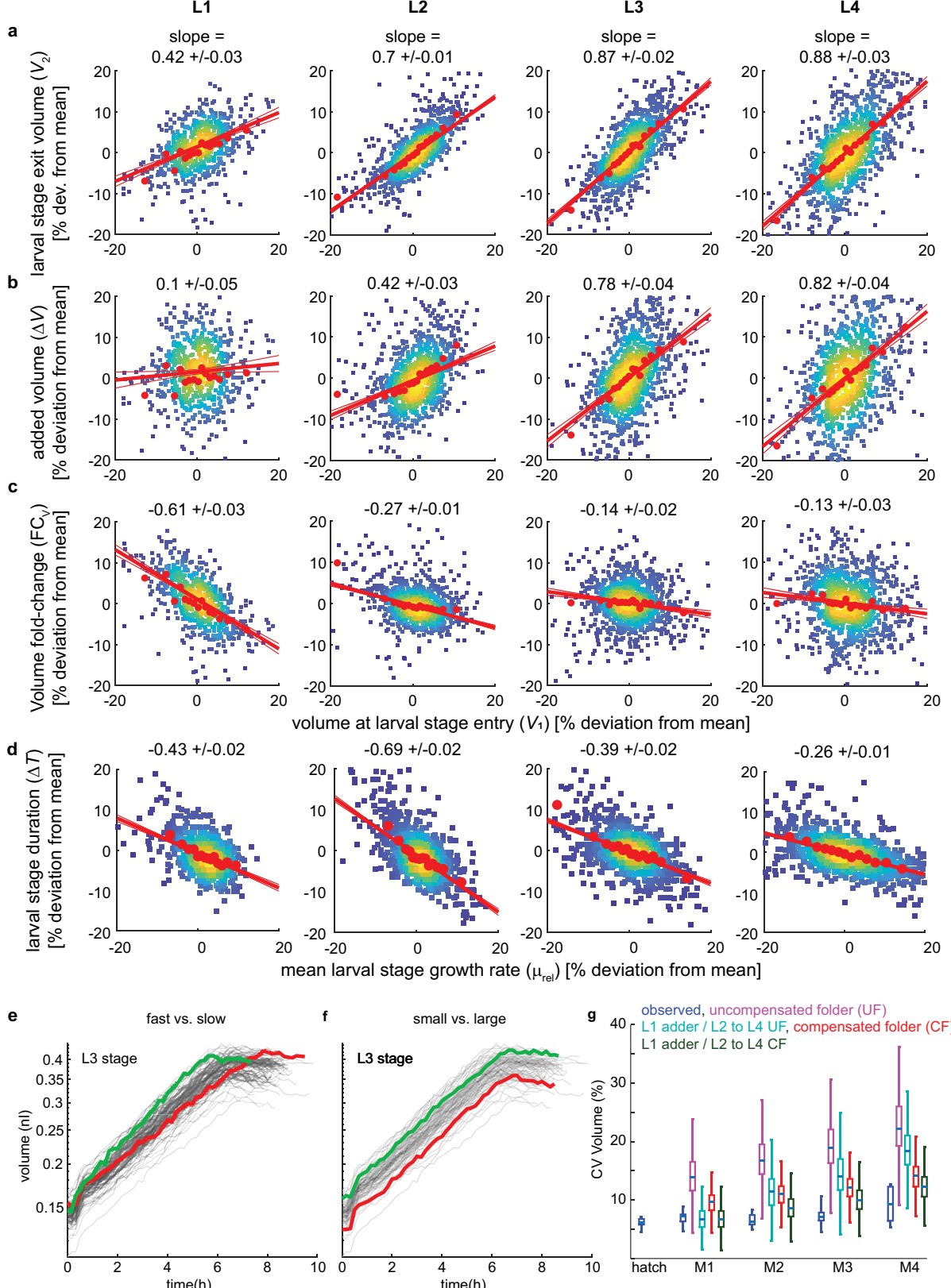

**e** fast vs. slow

**f** small vs. large

**g** observed, uncompensated folder (UF)
L1 adder / L2 to L4 UF, compensated folder (CF)
L1 adder / L2 to L4 CF

$\Delta V$ were also positively correlated with $V_1$ at 20 °C (Supplementary Fig. 6). We observed a slightly stronger negative relation between $V_1$ and $FC_V$ for L2 and L4 stages ($-0.69$, $-0.38$, $0.03$, $-0.35$ at 20 °C vs. $-0.61$, $-0.27$, $-0.14$, $-0.13$ at 25 °C for L1 to L4, Supplementary Fig. 6), indicating a weak size dependence of volume growth at lower temperature.

In summary, unlike for bacteria, yeasts, and cultured mammalian cells, we do not find adders and sizer mechanisms for *C. elegans* after the first larval stage. Instead, we find that the volume fold change per larval stage is nearly invariant with respect to the starting volume. In analogy to adders and sizers, we call this behaviour a folder (Fig. 3).

**Fig. 4 Near constant volume fold change per larval stage for L2 to L4. a** Scatter plot of volume at larval stage entry vs. volume at larval stage exit shown as % deviation from the mean for indicated larval stages. Colour indicates point density. Red dots are a moving average along *x*-axis ±SEM. Red trendline: robust linear regression to the data (see methods). Slope ±95% CI of the trendline is indicated above the panels. Thin red lines indicate 95% CI of the fitted model. **b** Same as (**a**), but for volume at larval stage entry vs. absolute volume increase per larval stage. **c** Same as (**a**), but for volume at larval stage entry vs. volume fold change per larval stage. **d** Same as (**a**), but for growth rate vs. larval stage duration. **e** Illustration of compensated folder. Grey lines: volume as a function of time of the L3 stage of all individuals of one experimental repeat. Two highlighted individuals (red and green) with similar starting volume, but different growth rates reach the same volume at the end of the larval stage. **f** Same as (**e**), but highlighting two individuals with similar growth rates, but distinct starting volumes that maintain the same relative volume difference at the start and end of the larval stage. **g** Comparison of CV of volume of four simulations (see text for details) with experimental observations as indicated. Blue box plot shows the CVs from $n = 10$ biologically independent experiments. Other colors show CVs from 1,000 simulations by randomized reshuffling as described in the text. p-value for model vs. data of CV (volume) at M4 (ANOVA, Tukey–Kramer multiple testing correction, two-sided): $<10^{-12}$, $3*10^{-12}$, 0.0039, 0.1583 (for purple, cyan, red, green). Boxplots: central line: median, box: interquartile ranges (IQR), whisker: ranges except extreme outliers (>1.5*IQR).

**Anti-correlation of growth rate and larval stage duration slows-down body size divergence**. A folder is expected whenever cells or organisms grow exponentially with time, and in a manner independent of their starting volume and current size (Fig. 3, Supplementary Fig. 7). Unlike adders and sizers, a folder is prone to volume divergence since stochastic deviations from the appropriate volume fold change are propagated and thus amplify during development (Supplementary Fig. 7a). We therefore asked why size divergence was nevertheless small during *C. elegans* development (Fig. 2).

Although a folder does not converge to a stable size distribution, divergence of body size between rapidly and slowly growing individuals can be reduced by other mechanisms. For example, slowly growing individuals can reach the same volume fold change as a rapidly growing individuals if they grows for a longer amount of time (Supplementary Fig. 7b, c). Indeed, we observed an anti-correlation between the growth rate $\mu$ and the larval stage duration $\Delta T$ (Fig. 4d, $R = -0.71, -0.75, -0.66, -0.64$ for L1 to L4, $p < 10^{-50}$ for all stages), consistent with previous measurements of tens of individuals[5]. We refer to the observed mechanism as a "folder with growth rate compensation". Unlike for sizers and adders, for a compensated folder, individuals with small starting volumes do not catch-up in size (Fig. 4f). Nevertheless, individuals with slow growth rates undergo the same volume fold change as rapidly growing individuals (Fig. 4e).

**The adder during L1 stage and the anti-coupling of growth and larval stage duration are required to counteract body volume divergence**. We have shown above that unlike L2 to L4 stages, the L1 stage uniquely follows an adder mechanism (Fig. 4b). To evaluate the relative contribution of this L1 adder and the compensated folder during L2 to L4 to body size uniformity, we compared the experimentally observed volume divergence with the volume divergence resulting from simulations of four different mechanisms: (i) complete uncoupling between growth and the duration of larval stages (uncompensated folder), (ii) an adder during L1, followed by an uncompensated folder during L2 to L4, (iii) a compensated folder throughout development, and (iv) an adder during L1, followed by a compensated folder during L2 to L4. In these simulations, the volume divergence continuously decreased from (i) to (iv) (Fig. 4g). The simulations thus show that neither growth rate compensation alone (iii), nor L1 adder alone (ii) are sufficient to approximate experimental observations and that both mechanisms are of functional importance. The experimentally observed volume divergence is even slightly smaller than the simulation of an L1 adder plus compensated folder (Fig. 4g), consistent with the weak deviation of experimental measurements from a perfect folder during L2 to L4 discussed above (Fig. 4c).

In summary, our temporally resolved growth measurements of individual animals revealed five characteristics of *C. elegans*

growth: (1) There is significant heterogeneity in the growth rate among individuals that persists during development (Fig. 2). (2) The L1 stage behaves like an adder, similar to many unicellular systems[12–17,19–22] (Fig. 4b). (3) From L2 onwards, larval stage transitions do not involve strong size thresholds, such as sizers or adders (Fig. 4c). (4) An inverse coupling between the rate of growth and the duration of development (compensated folder) reduces body size divergence despite the apparent absence of strong threshold-based mechanisms (Fig. 4d). (5) adder during L1, and compensated folder from L2 to L4 are both needed to maintain body size uniformity (Fig. 4g).

**Coupling of growth and development is robust to changes in growth rates**. Since body size homeostasis relies on an inverse coupling between the rates of growth and the duration of development, we asked if the coupling of growth and development was impaired by genetic manipulation of pathways known to control growth and/or developmental timing. First, we used a mutation of *eat-2*, which impairs growth by a reduction of pharyngeal pumping, and thus of food intake[34]. Second, we impaired mTOR signalling by a deletion of the RagA homolog *raga-1*[35,36]. Third, we perturbed TGFβ signalling by over-expressing the TGFβ ligand DBL-1[37] and by a mutation of the TGFβ target *lon-1*[3,38]. Fourth, we perturbed developmental timing by a mutation in the heterochronic gene *lin-14*[39].

To compare these mutants with the wild type, we define the rate of development $\alpha$ as the inverse of the larval stage duration ($\alpha = 1/\Delta T$). Mutations with proportional effects on $\alpha$ and on the growth rate $\mu$ do not alter the volume fold change ($\ln(\text{FC}_V) = \ln\left(\frac{V_2}{V_1}\right) = \mu \Delta T = \frac{\mu}{\alpha}$). Differences in the body volume among mutants can therefore stem from non-proportional changes to $\mu$ and $\alpha$, or from differences in the volume at hatch.

Mutants differed only weakly from the wild type in their volume at hatch (Fig. 5b), but they differed substantially among each other and from the wild type in $\mu$ and $\alpha$ (Fig. 5a). Overall, mutant effects on $\mu$ and $\alpha$ were positively correlated (Fig. 5a, $R = 0.91$ [95% CI: 0.41, 0.991], 0.89 [0.28, 0.99], 0.63 [−0.36, 0.95], 0.95 [0.45, 1.00] for L1 to L4), indicating a coupling of growth and development across different genotypes. However, individual mutants deviated from perfect proportionality between $\alpha$ and $\mu$, resulting in significant alterations in body volume (Fig. 5b). For example, mutation of *eat-2* consistently slowed down the growth rate more strongly than the rate of development (Fig. 5a), resulting in animals that were smaller than wild type animals of the same larval stage (Fig. 5b). *lon-1* mutants were near proportionally affected in volume growth and developmental rate and did not change in final volume (Fig. 5b), although they were significantly longer and thinner than wild type animals (Fig. 5c)[40]. Finally, *lin-14* mutation reduced $\mu$ less strongly than $\alpha$, such that *lin-14* mutants were larger than wild type animals after three larval

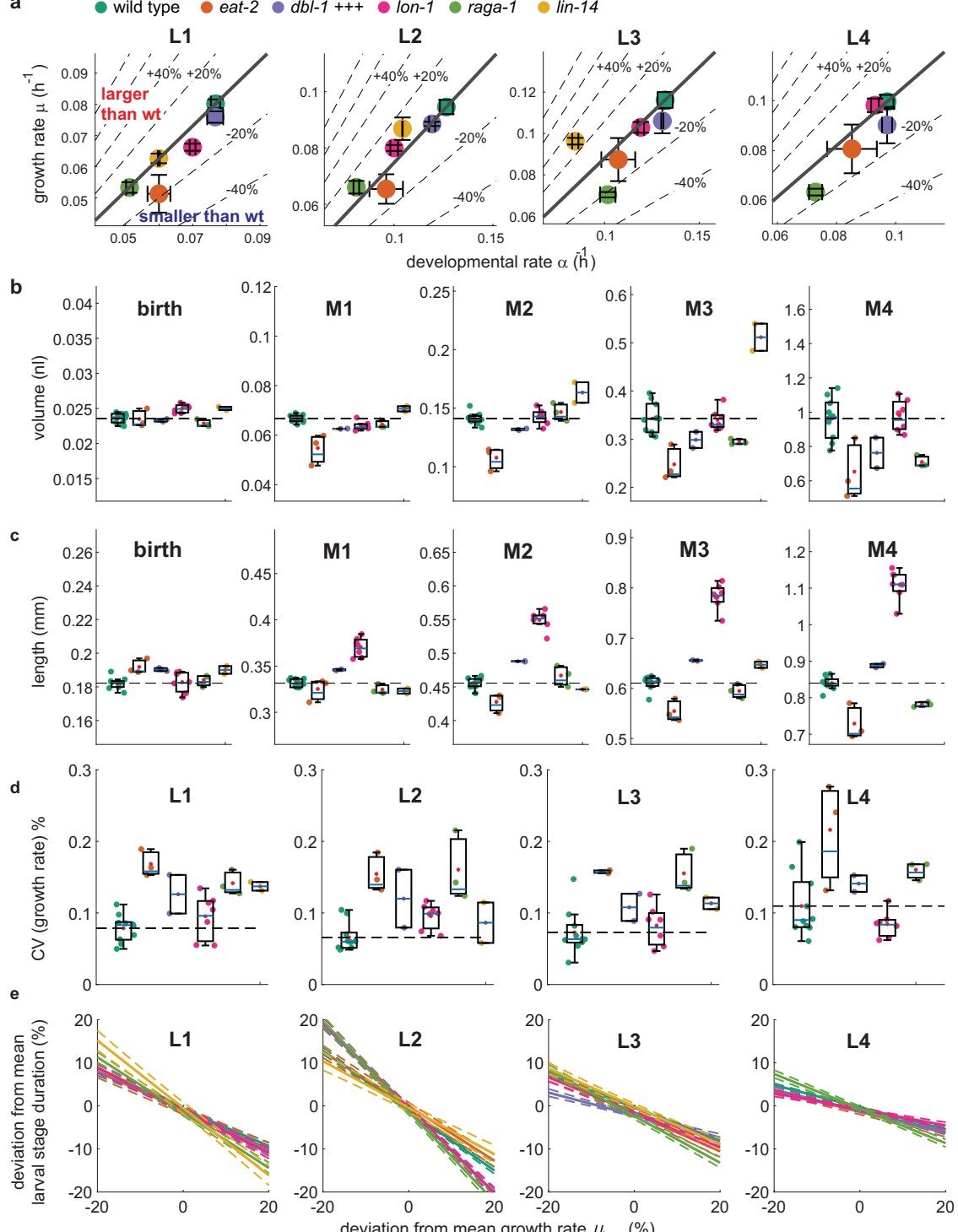

**Fig. 5 Mutations that alter growth and body size do not uncouple growth and development. a** Scatter plot of the rate of development $\alpha$ (=1/larval stage duration) vs. the volume growth rate for indicated mutants and larval stages. Error bars: SEM between day-to-day repeats. Thick grey line indicates proportional scaling corresponding to the volume fold change of the wild type. Dotted lines indicate deviation in volume fold change for a given deviation from proportionality. Region above the thick line corresponds to an increase in volume fold change, region below the thick line to a decrease in volume fold change compared to the wild type. **b** Box plot of volumes at birth and larval moults for indicated mutant strains. Colour scheme as indicated in legend of (**a**). Individual points and box plots indicate average volume of each day-to-day repeat. For p-values calculated by two-sided ranksum test of day-to-day repeats between mutant and wild type see Supplementary Table 1. **c** Same as (**b**), but for body length. **d** Same as (**a**), but for coefficient of variation of the growth rate. **e** Correlation between growth rate and the larval stage duration among individuals for different mutant backgrounds shown as deviation from the mean. Colours correspond to legend shown in (**a**). Solid lines show a robust linear regression to data. Dotted lines are 95% confidence interval of the fit. Scatter plot of individual data points is omitted for clarity of presentation (see Supplementary Fig. 8). **a–d** Boxplots: central line: median, box: interquartile ranges (IQR), whisker: ranges except extreme outliers (>1.5*IQR). Number of animals (n) for L1 to L4 from number of independent experiments (m): wild type: $n = 639$, 1,142, 1,144, 1,095; $m = 10$. *eat-2*: $n = 270$, 439, 440, 417; $m = 3$. *db++*: $n = 433$, 643, 650, 640; $m = 2$. *lon-1*: $n = 450$, 712, 716, 701; $m = 7$. *raga-1*: $n = 347$, 504, 501, 483; $m = 3$. *lin-14*: $n = 288$, 410, 387, n/a. $m = 2$.

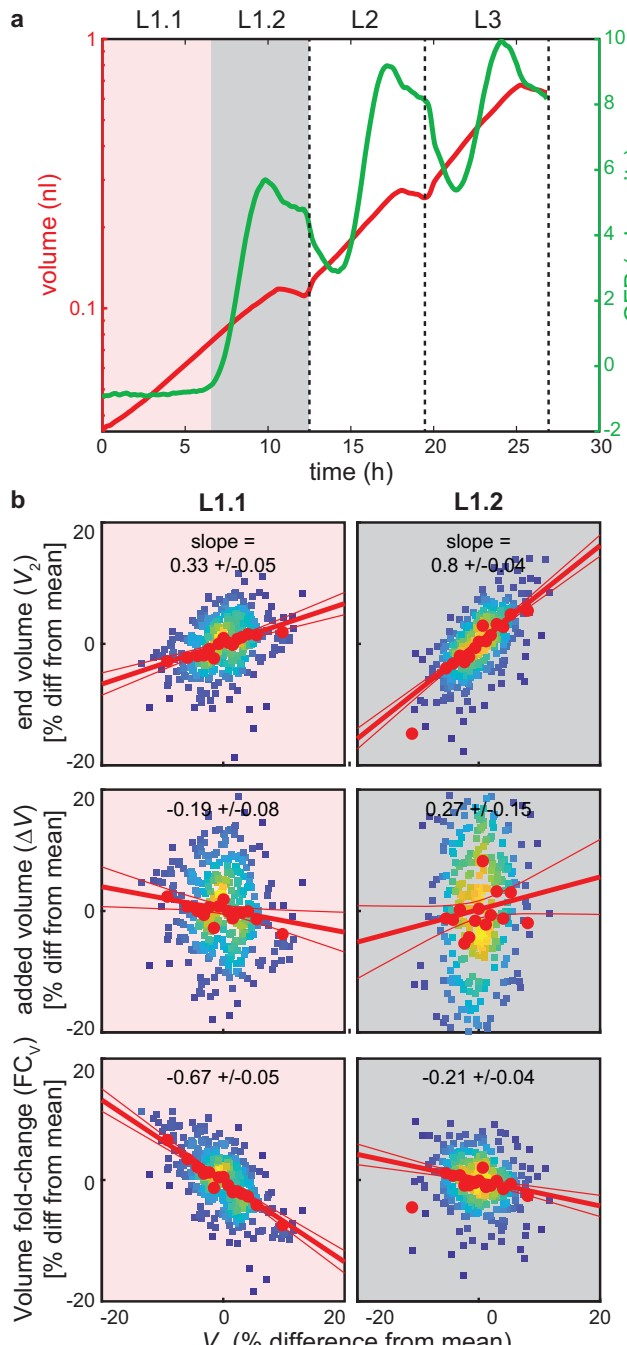

**Fig. 6 Folder coincides with developmental oscillations. a** Measurement of GFP concentration (green) and volume (red) of *dpy-9p::gfp* strain for larval stages L1 to L3. L4 stage was not measured in this experiment due to technical constraints. Figure shows GFP concentration and volume averaged over all individuals after temporal re-scaling of each stage separately to align individuals with slightly different growth rates. After re-scaling and averaging, the data was scaled back to the average larval stage duration. L1 is split into pre-oscillation (red, L1.1) and post-oscillation (grey, L1.2) substages. **b** Adder and sizer comparison for L1.1 and L1.2 as described in Fig. 4.

stages. Since *lin-14* mutant animals undergo only three larval stages[39], *lin-14* mutants were nevertheless smaller than the wild type at transition to adulthood (Fig. 5b). The increase in volume fold change for *lin-14* mutants was most prominent in L3 and was not observed during L1, consistent with our notion that size

control during L1 follows rules distinct from those later in development (Fig. 5a). Together, these data show that genetic mutations can affect the rate of growth and the rate of development non-proportionally, which alters the adult body volume.

We next asked if a non-proportional change of $\alpha$ and $\mu$ also disrupted the anti-correlation of growth and development among different individuals of the same genotype. For all mutant strains, the growth rate and the duration of larval stages remained anti-correlated with a quantitative relationship close to that of wild type animals (Fig. 5e, Supplementary Fig. 8, Supplementary Tables 2 and 3). Although several mutants had significantly increased heterogeneity in their growth rate (Fig. 5d), the volume divergence of mutants remained much smaller than expected by randomized simulations and was only slightly smaller than expected by the compensated folder model (Supplementary Fig. 9), like we observed for wild type. An exception to this rule was the *eat-2* mutant, for which divergence was close to expectations from random shuffling from the L2 stage onwards (Supplementary Fig. 9). We cannot distinguish if the increased heterogeneity among *eat-2* mutant individuals is a general effect of dietary restriction, or a characteristic of this specific mutant.

In summary, we conclude that the coupling of growth and development among individuals is robust to impairment of growth rate uniformity and is independent of the pathways investigated (Fig. 5e, Supplementary Figs. 8 and 9, Supplementary Tables 2 and 3). Quantitative differences in the volume divergence among mutants may relate to heterogeneities in traits that cannot be determined by volume measurements alone.

**The folder mechanism temporally coincides with the onset of developmental oscillations.** Since perturbation of canonical growth control pathways did not impair the coupling of growth and development, we asked if such coupling could be an intrinsic property of the oscillatory clock that times *C. elegans* development[23,24]. To test this, we first focused on the L1 stage which, unlike other larval stages, follows an adder and not a folder mechanism (Fig. 4). The L1 stage also differs from other larval stages with respect to the developmental oscillator: During L2 to L4 stages, oscillations occur in synchrony with larval stages, whereas the oscillator is arrested during the first 5–7 h of development[23] (Fig. 6a). We therefore asked whether the transition to the folder mechanism temporally coincided with the onset of gene expression oscillations.

To test this, we imaged growth of animals expressing *gfp* under control of the oscillatory collagen promoter *dpy-9p*[23] (Fig. 6a). As expected, *dpy-9p::gfp* expression was undetectable during the first 6 h of the L1 stage when the oscillator is arrested[23], and *gfp* expression oscillated in synchrony with larval stages later in development (Fig. 6a). Using *dpy-9p::gfp* expression, we divided the L1 stage into two substages (L1.1 and L1.2) before and after oscillations start. L1.1 behaved close to an adder (Fig. 6b), similar to what we observed for the entire L1 stage (Fig. 4b). In contrast, L1.2 was close to a folder (Fig. 6b), similar to stages L2 to L4 (Fig. 4c). Hence, the folder mechanism occurs specifically during the developmental window of active oscillations.

**Uncoupling the oscillatory frequency from the growth rate alters body volume.** To ask directly how developmental oscillations impact body volume, we next sought to alter the frequency of oscillations experimentally. Although the molecular mechanism driving oscillations remains poorly understood, recent evidence points to an important role of the transcription factor BLMP-1/PRDM1[41,42]. BLMP-1 oscillates at mRNA and protein level[41] and *blmp-1* mutation and depletion affects the duration of

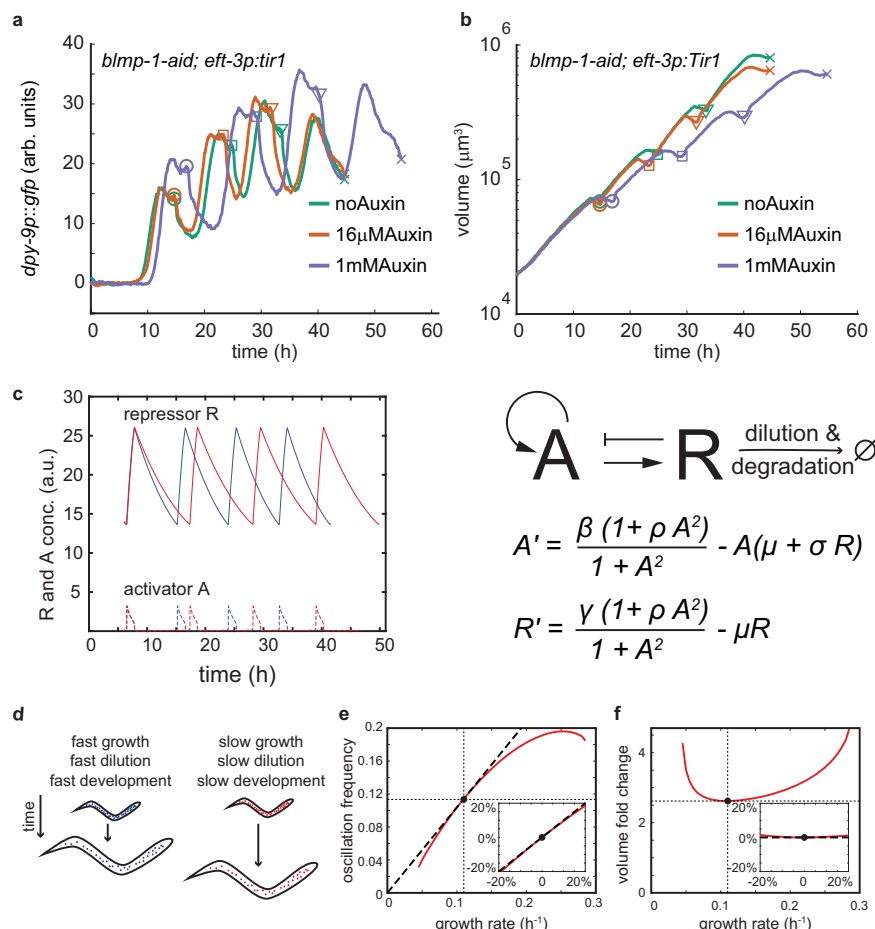

**Fig. 7 Changing the oscillatory frequency modulates body volume. a** *dpy-9::gfp* expression of strain with *blmp-1::aid*, *eft-3p::tir1* at indicated auxin concentrations. Averaging of individuals was conducted as described in Fig. 6a. GFP is shown as intensity per pixel. circle, square, triangle, and cross indicate M1-M4 in the respective condition. **b** Same as (**a**), but for volume. **c** Mathematical model of genetic oscillator based on design II of ref. [44]. Activator *A* activates its own production and the production of the repressor *R* by a factor of $\rho$. *R* degrades *A* at a rate $\sigma$. $\beta$ and $\gamma$ are the basal production rates of *A* and *R*, respectively. $\mu$ is the growth rate. *R* is considered stable, such that its removal rate is set by the growth rate $\mu$. Model parameters ($\beta = 10/$h, $\gamma = 0.3/$h, $\sigma = 10/$h, $\rho = 50$) are close to a saddle-node on invariant circle (SNIC) bifurcation[45] as is experimentally observed[23]. Modelled dynamics of *A* (dotted lines) and *R* (solid lines) for $\mu = 0.12/$h (blue), and for $\mu = 0.10/$h (red). **d** Impact of growth rate on the removal rate of a protein. In a rapidly growing individual (blue, left), a stable protein dilutes more quickly than in a slowly growing individual. The dilution rate is equivalent to the growth rate $\mu$ in the model (**c**). **e** Oscillation frequency as a function of the growth rate (red). Missing values at slow and fast growth rates are due to bifurcation to a non-oscillating state. Black dotted line: proportionality between the growth rate and the oscillation frequency, which would ensure perfect size homeostasis. Black circle: reference growth rate at which proportional scaling is precise for the given parameters ($\mu = 0.11/$h). Insert shows the same data but mean normalized to the reference growth rate ±20%, equivalent to the display of experimental observations shown in Figs. 4 and 6. **f** Same as (**e**), but for growth rate vs. volume fold change. The invariance of the volume fold change with respect to the growth rate is qualitatively independent on parameters over a wide range (see Supplementary Fig. 10).

larval stages. Interestingly, the effect of BLMP-1 depletion on larval stage duration is non-monotonic. Weak depletion of BLMP-1 shortens larval stages (particularly the intermolt), whereas strong depletion or null mutation of *blmp-1* extends the larval stage duration[41]. To ask how these effects on larval stage durations are related to potential changes in gene expression oscillations and/or the growth rate, we used a strain in which the *blmp-1* gene was endogenously engineered with an auxin inducible degradation (AID) tag[41,43]. We could thereby titrate the degradation rate of BLMP-1 using a high (1 mM) and a low (16 μM) dose of auxin in micro chambers, although due to the low basal expression level of BLMP-1 it was not possible to measure BLMP-1 levels directly.

Consistent with previous observations[41], larval stage durations were shortened by 16 μM auxin, and extended by 1 mM auxin (Supplementary Fig. 10a). The shortening of the larval stage duration by a low dose of auxin was particularly pronounced at

the L2 stage, for reasons that we have not explored further, but that could be due to technical or endogenous stage-specific modulation of BLMP-1 activity. At both auxin concentrations, changes in larval stage durations were matched by an appropriate temporal scaling of *dpy-9::gfp* oscillations (Fig. 7a), confirming a tight connection between oscillations and the rate of development[23]. Importantly, although 16 μM auxin accelerated *dpy-9::gfp* oscillations and shortened L2 duration by more than 15% (1.6 h), this low dose of auxin had nearly no effect on the growth rate (0.0031 h$^{-1}$, <3%). Consequently, this intervention uncoupled growth from development and reduced the body volume (Fig. 7b, Supplementary Fig. 10b). 1 mM auxin slowed down growth, presumably due to pleiotropic effects as was observed in *blmp-1(0)* mutants[41], which explains the observed developmental delay. Nevertheless, also at 1 mM auxin, the relation between growth and larval stage duration was perturbed and the body volume was significantly reduced (Fig. 7b and

Supplementary Fig. 10b). Together, these data suggest that growth and development can be uncoupled by the targeted acceleration of oscillations, supporting a role of oscillations in the control of larval stage duration and of body size.

**Coupling of growth and development is an emergent property of a genetic oscillator**. To better understand how developmental oscillations relate to volume growth, we analysed a classic mathematical model of a genetic oscillator (called here the "A/R model")[44] described by an ultrasensitive feedback between an activator protein A and a repressor protein R (Fig. 7c). Although in its entirety, the developmental clock of *C. elegans* is likely more complex, the A/R model captures many of its dynamic features, including its operation close to a saddle-node on invariant circle (SNIC) bifurcation[23,45]. In the A/R model, the oscillatory frequency scales positively with the removal rate of the repressor R over a wide parameter range[44] (Fig. 7c). The model thereby explains the acceleration of oscillations upon increased BLMP-1 turnover, if BLMP-1 fulfills a function analogous to the repressor R in the model (Fig. 7b).

In the context of a growing system, the removal rate of a protein in terms of its concentration is determined by the sum of its biochemical degradation rate and its dilution through growth[46]. Thus, if the degradation rate of R is small relative to the growth rate, in the model, the oscillatory frequency is expected to scale with the growth rate (Fig. 7d). Indeed, simulations show that for biochemically stable R, the oscillation frequency $\alpha$ scales near proportionally with the growth rate $\mu$ over a wide parameter range (Fig. 7e, and Supplementary Figs. 10c, d). The A/R model therefore suggests that the coupling of growth and development (as we observed experimentally) can emerge as an intrinsic property of a developmental oscillator without the need for additional complex control. Conversely, modulating the oscillatory frequency independent of the growth rate, e.g. by increasing the biochemical degradation rate of R, alters body volume, as we have observed experimentally for BLMP-1 (Fig. 7b).

## Discussion

Exponential growth of cells and organisms presents the challenge that small differences in the growth rate can, in principle, amplify to large differences in size. Unicellular systems overcome this challenge by adder and sizer mechanisms, where the volume fold change per cell cycle correlates negatively with the size at birth[12–17,20–22]. Although we found that the first larval stage of *C. elegans* follows an adder mechanism, for the rest of development size uniformity is not attained by strict size thresholds. Instead, size divergence is counteracted by an inverse coupling of the growth rate to the duration of development, such that rapidly growing individuals grow for a shorter amount of time and vice versa (Fig. 4).

The coupling of growth and development reduces volume divergence, but unlike adders and sizers does not entirely prevent the accumulation of size differences during development (Fig. 2). Yet, such perfection may not be required in the case of *C. elegans* development, which involves only four larval stages and ~5.5 volume doublings, compared to hundreds of cell divisions undergone by bacteria. Nevertheless, also in unicellular systems, size control mechanisms involving growth rate regulation have been observed, in addition to size-dependent scaling of the cell cycle duration[22,47,48].

The volume of L1 larvae after hatch is sensitive to external perturbations. For example, the size of L1 larvae depends on the age of their mothers[8], and extended starvation of L1 larvae after hatching reduces their volume[49,50]. The adder mechanism during the L1 stage may be important to counteract such size heterogeneities and to prevent their propagation to later stages of development. Consistently, the reduction of size by external perturbation correlates with an extended duration of the L1 stage[50,51]. This time delay may be required for animals to recover their appropriate volume. It will be interesting to investigate if this compensatory mechanism relates to the L1 adder and if it similarly involves a size-dependent onset of gene expression oscillations.

Our observations of coupled growth and development are consistent with the observation that the temporal spacing of different morphologically defined events of *C. elegans* development scale proportionally among individuals[52]. Our model further suggests that the time between developmental events is determined by the rate of exponential growth of an individual. We thereby propose the growth rate as a central regulator of organismal physiology and development, similar to growth laws found in bacteria[53,54]. Identification of such organismal growth laws provides a powerful framework for predictive analysis of multicellular physiology.

At least two distinct, but not mutually exclusive, mechanisms may couple growth and development. First, mechanical stretching of structural components, such as the cuticle, could trigger moulting in ways that maintain a constant volume fold change[55,56]. Alternatively, the growth rate may influence the dynamics of a developmental clock, such as the developmental oscillator of *C. elegans*[23,24]. According to this latter model, the coupling of growth and development is a continuous process that occurs throughout each larval stage, rather than a singular event occurring at the larval moult. Such continuous coupling is also consistent with proportional scaling of developmental events among individuals[52] and the temporal scaling of gene expression oscillations at reduced temperature[24].

Increased turnover of the oscillatory transcription factor BLMP-1 accelerated oscillations and uncoupled the rates of growth and development to produce animals of reduced body volume (Fig. 7a, b). The proposed A/R model, consisting of a two-component feedback circuit, explains this behaviour (Fig. 7) and suggests that an inverse coupling of growth and development could emerge as an intrinsic property of the oscillator. In additional support of a contribution of the developmental oscillator to body volume uniformity, we showed that the size independence of the volume fold change is restricted to the developmental window where the oscillator is active (Fig. 6a, b).

The oscillatory clock of *C. elegans* is likely more complex than described by the A/R-model. For example, although *blmp-1(0)* mutation impairs the synchrony of oscillations, this mutation does not disrupt oscillations entirely[41], suggesting that redundant, and potentially coupled, oscillators exist. Nevertheless, as for the A/R model, it is likely that even for more complex networks, the oscillatory frequency is impacted by dilution through growth, which intrinsically counteracts amplification of size heterogeneity. The modelling approach presented here therefore serves as a powerful framework for quantitative predictions of the role of other candidate factors and can guide a characterization of the entire network in future studies.

Transcriptional oscillations are found in numerous organisms. Most famously, circadian clocks control oscillations that match the diurnal cycle[57]. Unlike the developmental clock of *C. elegans*, the 24 h period of circadian clocks is robust to fluctuations in growth rates or temperature[57]. We propose that the apparent lack of robustness of developmental oscillations to changes in growth rates in return provides robustness to body size. It will be interesting to see if this design principle also applies to the size homeostasis of other multicellular systems, such as the somitogenesis clock in vertebrates[58].

## Methods

**Caenorhabditis elegans strains.** The following strains were used in this study:

HW1939: *xeSi296[eft-3p::luc::gfp::unc-54 3'UTR, unc-119(+)] II* ([23])

HW2688: *xeSi296[eft-3p::luc::gfp::unc-54 3'UTR, unc-119(+)] II; lon-1(e185) III.* (this study)

HW2696 *xeSi301[eft-3p::luc::gfp::unc-54 3'UTR, unc-119(+)] III.; raga-1(ok386) II.* (this study)

HW2687: *xeSi296[eft-3p::luc::gfp::unc-54 3'UTR, unc-119(+)] II; lon-1(e185) III. ctIs40[dbl-1(+) sur-5::GFP] X.* (this study)

HW2681: *eat-2(ad1113) xeSi296 [eft-3p::luc::gfp::unc-54 3'UTR, unc-119(+)] II.* (this study)

HW1973: *xeSi296 [eft-3p::luc::gfp::unc-54 3'UTR, unc-119(+)] II.; lin-14 (n179)* (this study)

HW2840: *xeSi449[eft-3p:mCherry-luciferase unc-119(+)] III.; xeSi440[dpy-9p::GFP::H2B::Pest:: unc-54 3'UTR; unc-119(+)] II.* (this study)

WBT241: *blmp-1(xe80 [blmp-1::AID]) I; xeSi440[dpy-9p::GFP::H2B::Pest::unc-54 3'UTR, cb-unc-119 (+)] II; xeSi376[eft-3p::TIR1::mRuby::unc-54 3'UTR, cb-unc-119(+)] III; wbmIs88[eft-3p::3xFLAG::dpy-10 crRNA::SL2::wrmScarlet::unc-54 3'UTR] V:8645000* (this study)

The GFP reporter *xeSi301* is equivalent to *xeSi296*[23], except that it was inserted on chromosome III instead of chromosome II. *xeSi449* is the same reporter as *xeSi301*, with mCherry instead of GFP. All transgenes were inserted by MosSCI as described in[59], except for *wbmIs88*, which is a single copy insert created using CRISPR described in ref. [60], and *ctIs40* which is an integrated multi-copy array.

**Live imaging in micro chambers.** Imaging of individual animals was performed using a protocol adapted from[26] and as described in detail in[23] except that a 3.5 cm dish with optical quality gas-permeable polymer (ibidi) was used to mount the chambers. In brief, arrayed micro chambers were produced from a 4.5% Agarose gel in S-basal using a PDMS template as a micro comb to create the chambers. Dimensions of chambers used in all Figures except Fig. 6a, b were $600 \times 600 \times 20\,\mu m$. Chambers used in Fig. 6a, b were $370 \times 370 \times 15\,\mu m$ for compatibility with camera chip size of the microscope used for these experiments. Chambers were filled with bacteria of the strain OP50 (or OP50-1 for Fig. 6e, f, S3, S6), which was scraped off a standard NGM plate using a piece of 3% agar in NGM, supported by a 25 mm x 75 mm glass slide. After placing individual eggs into the chambers, the microchamber arrays were inverted onto an ibidi dish for imaging. The remaining space around the dish was filled with 3% low melting temperature agarose dissolved in S-basal (cooled down to 42 °C prior to application to the dish). The dish was sealed with parafilm to prevent desiccation of the agarose during imaging. For experiments in Fig. 5a, b and Supplementary Figs. 3 and 6, the agarose was additionally overlayed with ~0.5 ml of PDMS which was allowed to cure during the acquisition and applied to minimize condensation on the lid and drying out of the sample.

For all experiments except those in Fig. 6, a 10x objective was used on an Olympus IX70 wide-field microscope and an sCMOS camera with a pixel size of 6.5 µm and 2 × 2 binning. At each timepoint, a z-stack of 7 planes with 5 µm spacing was acquired using a piezo-controlled stage using a 10 ms exposure time. The focal plane with best contrast was automatically selected for further image analysis. Experiments in Fig. 6 were conducted on a Yokogawa spinning disc microscope equipped with two EM-CCD cameras and a beam-splitter. GFP and mCherry signals were acquired sequentially with a total time delay of 35 ms. This delay was sufficiently short to overlay the two channels without substantial movement of the animal. At each timepoint, a z-stack of 35 µm with 5 µm z-spacing was acquired. The volume at each timepoint was computed from the central focal plane. The fluorescence was computed from the sum of all planes divided by the total number of pixels. For each experiment, control animals without a GFP reporter (HW2840) were imaged and fluorescence of developmental stage-matched animals was used to subtract background and autofluorescence of GFP. Experiments shown in Fig. 7a, b were conducted on a Nikon Ti2 wide-field epifluorescence microscope with a 10x, 0.45 NA objective. Software auto-focus was used to find the central focal plane in the mCherry channel marking the body and a single image was taken in GFP and mCherry using single band pass filters. Segmentation was conducted as described above on the mCherry signal to compute the size (number of pixels n) of each individual at each time point. To compute the total GFP intensity, the n brightest pixels were summed and the mean of lower 50% percentile of pixels of the entire frame was used for background subtraction. This procedure was chosen due to the time delay between mCherry and GFP acquisition caused by changing of the filters, which was not compatible with overlaying the two channels. Background and autofluorescence were subtracted using the average intensity measured in worms during the the first half of the L1 stage, where the reporter is inactive. For all experiments, the temperature was maintained at 25 °C or 20 °C using an incubator encapsulating the entire microscope (life imaging services).

Auxin (IAA, Sigma) solutions were freshly prepared on the day of the experiment as a 400x stock in EtOH and subsequently diluted to the indicated concentration in agarose to a final EtOH concentration of 0.25% immediately prior to use for micro chamber assembly.

**Image analysis.** A custom Matlab script was used to segment worms from raw images. The ImageJ "straightening function" embedded in a KNIME workflow[61] was used for straightening. For segmentation, edge detection by Sobel algorithm using the *edge()* function of Matlab was used, followed by connecting endpoints closest to each other to close gaps in the detected outline. After straightening, each image was classified as either as 'egg', or 'worm' using a decision tree based classifier that was trained on a small subset of manually assigned images. This classification also identified cases where straightening failed (e.g. in the case of self-touching animals), which were removed from further analysis. Details of classifier: The following features were computed for every segmented and straightened image: length, standard deviation of width, cv of width, maximal width, median width, maximal width/median width, volume, volume/length, entropy of width. These features were then used to train an ensemble of 20 bagged (standing for "bootstrap aggregation") decision trees using the *TreeBagger()* function of Matlab. For classification of new data, the same features were calculated, and images were classified using the trained ensemble of decision trees.

**Computation of volume and detection of moults.** At each timepoint, the volume was computed from straightened images assuming rotational symmetry. The assumption of rotational symmetry is well justified, based on previous measurements[11]. Timepoints of larval stage transition were determined by the maximum of the second derivative of the logarithm of the volume. Each volume trace was subsequently inspected and curated manually using a custom-made graphical user interface written in Matlab. To minimize measurement errors, the larval volume at each moult was computed by a linear regression of the volume from ten timepoints preceding the moult (for M1 to M4) or regression to 10 timepoints after hatching for the volume at birth.

**Computation of growth rates.** To calculate the continuous linear and absolute growth rates shown in Fig. 1d, volumes were median filtered with a window of three time points and smoothed over 15 timepoints using the smooth function with rlowess option of Matlab. Individuals were then re-scaled to the duration of the larval stage and averaged after linear interpolation of the signal at 100 points per larval stage.

To calculate average growth rates per larval stage (Fig. 1d) a linear regression of time vs. ln(volume) was performed including all timepoints of a larval stage except the first 10% and the last 25% of each larval stage to avoid confounding effects of lethargus. Absolute growth rates were determined by the same procedure, but by a regression to the volume without log transformation.

**Normalization of day-to-day repeats.** Where shown as % deviation from the mean, growth rates, volumes, and times were normalized to the mean of each day prior to merging different days. The coefficient of variation was computed as the standard deviation divided by the mean of the normalized data.

**Simulation of randomized populations.** To determine the expected volume divergence in the absence of coupling of growth and development given the observed heterogeneity in growth and larval stage duration, simulations were carried out with a starting population of the measured body volumes at start. Each individual was then assigned a growth rate randomly drawn from the measured growth rates (determined as $\Delta \ln(V)/\Delta t$ to avoid confounding effects of lethargus) and a larval stage duration $\Delta t$ randomly drawn from the measured larval stage durations. From these parameters, the volume at the following larval stage was computed and the process was repeated iteratively until the end of L4. For fair comparison with measured data given effects of day-to-day variation, randomizations were performed for each day-to-day repeat separately. For each simulation, the number of simulated individuals was equal to the number of individuals in the measured data, and randomization was performed 1000 times for each day-to-day repeat. Box plots in Figs. 2d, 4g and Supplementary Figs. 2e, 8 and 10 show the distribution of the coefficients of variation of all simulations. An equivalent procedure was applied to simulate the combined adder/folder model, but instead, individuals were assigned $\Delta V$ (for L1) and $FC_V$ (for L2 to L4) randomly drawn from the measured data. To compute the randomized and folder model data for M1 to M4 in Fig. 3d and Supplementary Fig. 7, the starting volume of the simulation were the measured volumes at M1.

**Computation of trendline in correlation between measured variables.** To determine the trendline between two measured variables a robust linear regression was performed using the robustfit() method of Matlab (v2021b) and default parameters to reduce sensitivity to outliers. Display of all scatter plots is restricted to the region from −20% to +20% for clarity of display. Few individuals were out of this range as apparent in Supplementary Fig. 5.

**Mathematical model of genetic oscillator.** To model oscillations, we built on a previously published model by Guantos and Poyatos[44]. The model describes the protein dynamics of an activator $A$ and a repressor $R$. $A$ activates its production, as well as the production of $R$ by a sigmoidal input function with a Hill coefficient of 2. $R$ accelerates degradation of $A$ by a factor $\sigma$. The model describes protein concentrations of $A$ and $R$ assuming quasi steady-state for the respective mRNAs. In addition to active degradation of $A$ by $R$, $A$ and $R$ are diluted by growth at a rate

$\mu$. $\beta$ and $\gamma$ are the basal production rates of $A$ and $R$, and $\rho$ is the factor by which $A$ enhances the production of its target. Parameters of the model were: $\beta = 10/s$, $\gamma = 0.3/s$, $\sigma = 10$, $\rho = 50$. Conclusions were qualitatively robust to changes in these parameter values.

To calculate the oscillation frequency $\alpha$, the dynamics of $A$ and $R$ were numerically solved using Matlab for the range of values of $\mu$ where the system adopted limit cycle oscillations. Volume fold changes were computed as $FC_v = e^{\mu t} = e^{\mu/\alpha}$.

**Reporting summary**. Further information on research design is available in the Nature Research Reporting Summary linked to this article.

## Data availability

The size, growth, and fluorescence data generated in this study are provided in the Supplementary Information and Source Data file. Source data are provided with this paper.

## Code availability

The custom made Matlab software for image segmentation and volume measurement is available on https://github.com/btowbin/NatComm2022. FMI-specific Fiji plugin for straightening in ImageJ/KNIME is available here: fmi-ij2-plugins, https://zenodo.org/record/3560533#.Yhf2GejML8A

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

## Acknowledgements

We thank Iskra Katic for help in generating transgenic strains; Laurent Gelman, Stephen Bourke, and Jan Eglinger for help with imaging and image analysis; Gregory Roth for helpful discussions. We are thankful to Yannick Hauser for sharing data and reagents for the experiments on *blmp-1* prior to peer reviewed publication. We acknowledge support by the Microscopy Imaging Center at the University of Bern. B.D.T was a recipient of an HFSP LTF (000309/2013), a Marie-Curie IF (#751878), and an Engelhorn-Traudl Foundation fellowship. This work received funding from the Swiss National Science Foundation (SNSF) in the form of an Eccellenza Professorial Fellowship (PCEFP3_181204) to B.D.T., from the Novartis Foundation for Medical-Biological Research (Grant #20A011) and from the European Research Council (ERC) under the European Union's Horizon 2020 research and innovation program (Grant agreement No. 741269, to H.G.). The FMI is core-funded by the Novartis Research Foundation. Some strains were provided by the CGC, which is funded by NIH Office of Research Infrastructure Programs (P40 OD010440).

## Author contributions

B.D.T. performed all experiments except for Fig. 7a, b and Supplementary Figs 3 and 6, which were performed by K.S., B.D.T performed all data analysis, and modelling and wrote the manuscript. H.G. and B.D.T. jointly conceived the project. K.S. and H.G. edited the manuscript.

## Competing interests

The authors declare no competing interests.
