## [Peer Review File · Nature Communications]

Coupling of growth rate and developmental tempo reduces body size heterogeneity in *C. elegans*REVIEWER COMMENTS

Reviewer #1 (Remarks to the Author):

The authors explore the coordination between growth and development, which is a fundamental question in biology. They study this in *C.elegans* employing single animal live imaging to quantify both volume change and developmental progression, and mathematical modeling. First they found *C.elegans* mostly follows an exponential growth rate (except at the L4 stage) which might potentially amplify initial small differences in volume. Instead, they found the differences in volumes are small given heterogeneity in growth rate implying some control mechanism buffering variation volume variation. They found that growth control is different between the first and subsequent larval stages: while volume change during first larval stage is independent of initial volume, consistent with an adder mechanism, for the next larval stage they found that it is the fold change in volume that is independent of initial volume, a novel mechanism they term folder. This mechanism however would lead to extensive body size divergence during development which contrasts with observed low volume variation, implying some compensation mechanism. They found that *C. elegans* achieve this by adjusting the duration of larval stage to the growth rate: slower growing individuals develop for longer reducing volume variation among individuals. They found this inverse coupling between growth and development is robust to genetic perturbation of several pathways controlling growth or developmental timing. Finally, they observe that the change between the adder and folder mechanism coincides with the onset of oscillatory gene expression during first larval stage. They further show that the observed coupling between growth and development can be achieved without the need of complex regulation by a simple model involving a clock mechanism produced by feedback between an activator and a repressor where the repressor is diluted by growth. The model is purely speculative and no experimental evidence is provided to support it.

Overall, I think the findings are interesting and well supported by the large amount of data collected and properly analysed, the manuscript is clearly written and deserve publication.

I only have minor comments.

- I think the coincidence of the coupling between the onset of oscillations and the change between the adder and folder growth control is highly suggestive but it doesn't necessarily imply a causal link between the two as it suggested in the abstract.
- *lin-14* which is a heterochronic mutant slows down development more than producing worms larger than wt but it does so only after the first larval stage (after the oscillations kick-in in the wt). I think this is particularly interesting and may strengthen their point although the authors do not comment on that. Also, *Lin14* is a precocious mutant (i.e. undergo L2 specific events during the L1) so I think it would be interesting to see how the oscillations behave in this mutant and if they are consistent with their model like the wt.
- I don't see the point of computing trend lines on binned data instead of using the raw data. I think showing a trend line and relative confidence intervals computed on raw data together with the scatterplot should be clearer. It should be especially clearer in plots of figure 4e.

In the methods

- Image analysis paragraph is lacking some info: KNIME workflow... please provide a citation for KNIME. Random forest classifier, no detail is provided about the classifier.
- Figure S6 is missing annotation of the larval stages
- It's mainly a matter of taste but, to improve readability (and broaden the readership) I would avoid 'folder mechanism' in the title because it's a bit jargony and it is introduced for the first time by the authors in this manuscript.

Reviewer #2 (Remarks to the Author):

Towbin and Grosshans presents their results on the regulation of size uniformity during *C. elegans* development. They investigate if there is a working size control which ensures that at each larvae stage *C. elegans* have a more or less fixed size. The growth measurement and analysis have been carefully done, the finding of a new size controlling "folder" mechanism is relevant and in general the manuscript is nicely written. Although there are a few major issues that needs to be resolved before it could be published:

The general connection between size control in single cell eukaryotes and the reduced noise in size at developmental stages is interesting and indeed in L2-L4 the growth is more or less equal to the initial size, which is similar to single celled organisms, but in L1 the growth is much larger than the initial size, which already gives a hint that L1 will be differently handled than other stages. These actual changes can be seen only from Fig S4, while Fig S3 is explaining how such classical size control measurement can be converted to the % deviation scaled plots of Fig. 3. This Fig S3 is really important to understand key points, it could be even moved to the main text. A missing panel from this is one where volume added is given on the ordinate (like on all other figs). Here a -1 slope would mean a strong sizer (based on figS4 L1 has a weak sizer, which would compensate size difference in 5 cycles). This brings the next point: the cell cycle is a periodic process, so small changes in one cycle can be compensated in more than one cycles. In the developmental process discussed here, the oscillations could come from the developmental oscillations, but the number of these is quite limited. It would be great to hear why the compensation should be at each larval stage not only at a single step. The separate control on L1 suggest this as the crucial step, the folder mechanisms at later stages might be able to work only if coupled with a more precise adder or sizer step. I would like to see the calculation in the style of fig S3 to check how the folder model could keep homeostasis.

The mathematical model with a combined positive and negative feedback and how its period is driven by a growth rate resembling parameter nicely adds to the results, but it is not really surprising that the frequency of oscillations can be controlled by a parameter which jointly controls the half-life of both species. This parameter can scale the dynamics of the whole system in this way, it is just not fully clear why growth rate should act such way on the developmental oscillator.

The changes between growth and no-growth stages were also observed in old experiments on fission yeast cells (Sveiczzer Journal of cell science 1996 or Fantes Journal of cell science 1977). Interestingly a growth rate change was also observed in these cycles as well.

In summary, my main concern is: how far the size control of cell cycle can be correlated with the 4 stages of *C. elegans* developmental program. It was clearly shown that some size control mechanism works, but I am not fully convinced if all 4 stages have a critical role in size homeostasis and if the proposed folder model could indeed serve as a critical size regulator.

Reviewer #3 (Remarks to the Author):

This manuscript by Towbin and Grosshans describes a detailed examination of growth and body size across development in *C. elegans*. The methodology is strong and analyses solid, and the conclusions made are robust based upon state-of-the-art single animal imaging that is innovative. Much of the manuscript is spent characterizing wild type animals and accommodations for heterogeneity. A general weakness of the manuscript is that it is largely descriptive and only at the very end begins to get into the genetic underpinnings of body volume and growth rate control. I think the work is well conducted but is missing more detailed mechanism and thus based upon its more descriptive nature may be better suited for a specialty journal without the wide readership of Nature Communications.

The analyses are done uniformly at one temperature, and one at which many believe *elegans* is under heat stress. The authors should give thought to the effect of temperature on the process and whether the same relationships hold true.

The heterogeneity among the nutritional model used, *eat-2* mutants, is far greater than the other genetic mutants. Whether this is an intrinsic property of this mutant or of the effects of nutrient deprivation on the "oscillator" as the authors propose for wild type animals, could be tested by using diets of different "quality" e.g. as in MacNeil et al Cell 2013.

The testing of a heterochronic mutant is a nice touch, but in general the loss of machinery responsible for cyclic gene expression (oscillatory) is not explored, realizing the limitation that we do not fully understand the machinery necessary for oscillations in the worm. None the less, circadian gene expression has been described in detail in the worm (Goya et al. PNAS 2016; Benerjee et al Dev Cell 2005; review in Hasegawa et al Chronobiol Int. 2005), thus the possibility of entrainable oscillations and rhythms that the authors could explore mechanistically in more detail. The major question as to whether the oscillations are governed by growth or whether the oscillations govern growth is a critical one to answer. Presumably there will be oscillatory genes/proteins in both classes.

It may be possible to use genes/proteins that are expressed uniquely at the molt (e.g. *mlt-10*, *mlt-12*) to test the specific features of the folder hypothesis that the authors put forward. By their rationale, the expression of these proteins should only appear when the criteria specified by a folder mechanism are met.

Point-by-point response to the reviewers

Reviewer #1 (Remarks to the Author):

The authors explore the coordination between growth and development, which is a fundamental question in biology. They study this in *C.elegans* employing single animal live imaging to quantify both volume change and developmental progression, and mathematical modeling. First they found *C.elegans* mostly follows an exponential growth rate (except at the L4 stage) which might potentially amplify initial small differences in volume. Instead, they found the differences in volumes are small given heterogeneity in growth rate implying some control mechanism buffering volume variation. They found that growth control is different between the first and subsequent larval stages: while volume change during first larval stage is independent of initial volume, consistent with an adder mechanism, for the next larval stage they found that it is the fold change in volume that is independent of initial volume, a novel mechanism they term folder. This mechanism however would lead to extensive body size divergence during development which contrasts with observed low volume variation, implying some compensation mechanism. They found that *C. elegans* achieve this by adjusting the duration of larval stage to the growth rate: slower growing individuals develop for longer reducing volume variation among individuals. They found this inverse coupling between growth and development is robust to genetic perturbation of several pathways controlling growth or developmental timing. Finally, they observe that the change between the adder and folder mechanism coincides with the onset of oscillatory gene expression during first larval stage. They further show that the observed coupling between growth and development can be achieved without the need of complex regulation by a simple model involving a clock mechanism produced by feedback between an activator and a repressor where the repressor is diluted by growth. The model is purely speculative and no experimental evidence is provided to support it.

Overall, I think the findings are interesting and well supported by the large amount of data collected and properly analysed, the manuscript is clearly written and deserve publication.

We thank this reviewer for the endorsement of our work and address his/her minor comments as detailed below.

I only have minor comments.

- I think the coincidence of the coupling between the onset of oscillations and the change between the adder and folder growth control is highly suggestive but it doesn't necessarily imply a causal link between the two as it suggested in the abstract.

We agree with the reviewer's comment on lack of evidence supporting causality. Hence, and in response to a comment by Reviewer #3, we have now tested specifically whether uncoupling the oscillatory frequency from the growth rate would alter body volume, which we find to be the case. This supports a causal connection between oscillations and body size. We have revised the abstract to be more specific in our claims.

"We perturb this coupling by changing the developmental tempo through manipulation of a transcriptional oscillator that controls the duration of larval development. As predicted by a mathematical model, this perturbation alters the body volume."

- *lin-14* which is a heterochronic mutant slows down development more than producing worms larger than wt but it does so only after the first larval stage (after the oscillations kick-in in the wt). I think this is particularly interesting and may strengthen their point although the authors do not comment on that. Also, *Lin14* is a precocious mutant (i.e. undergo L2 specific events during the L1) so I think it

would be interesting to see how the oscillations behave in this mutant and if they are consistent with their model like the wt.

We thank the reviewer for pointing this out. We now discuss this observation in the text as follows in line 276:

“The increase in volume fold change for *lin-14* mutants was most prominent in L3 and was not observed during L1, consistent with our notion that size control during L1 follows rules distinct from those later in development (Fig. 5a).”

We have conducted additional experiments using live imaging in micro chambers of a *lin-14* mutant carrying the oscillating reporter *dpy-9::gfp*. As shown in the figure below, we find that the arrest of oscillations is retained during the first hours of development even in a *lin-14* mutant (middle panel). Thus, unlike other L1 specific features, arrest of oscillations is not dependent on *lin-14*. Instead, oscillations start at nearly the same volume in *lin-14* mutants as in wild type animals (see bottom panel, which shows *dpy-9::gfp* expression as a function of volume). It will be interesting to identify the mechanism that arrests oscillations in the L1 stage, and the molecules that trigger the onset of oscillations at a given (added) volume. However, we believe this data would not add to the mechanistic understanding of the coupling of growth and development, which is the focus of our current manuscript. We have therefore not included these figures in the revised manuscript.

I don't see the point of computing trend lines on binned data instead of using the raw data. I think showing a trend line and relative confidence intervals computed on raw data together with the scatterplot should be clearer. It should be especially clearer in plots of figure 4e.

The presentation of trend lines has been adjusted as suggested in all figures. To ensure that the slopes are not dominated by rare outliers, a robust linear regression method was chosen for this analysis. Specifically, we used an iteratively reweighted least-square algorithm implemented by the `robustfit()` function of Matlab (v2021b) with default parameters. This algorithm 'seeks to optimize fit to the bulk of the data while minimizing the effect of outliers', as detailed here: <https://ch.mathworks.com/help/stats/robust-regression-reduce-outlier-effects.html>.

While we believe this robust fit method is most appropriate for our purpose, we note that a standard least square fit (without correction for outliers) produces overall very similar results (see figure below).

robust least square

standard least square

- Image analysis paragraph is lacking some info: KNIME workflow... please provide a citation for KNIME. Random forest classifier, no detail is provided about the classifier.

The methods section was adjusted accordingly (line 694 ff.).

“A custom Matlab script was used to segment worms from raw images. The ImageJ ‘straightening function’ embedded in a KNIME workflow ⁶⁰ was used for straightening. For segmentation, edge detection by Sobel algorithm using the *edge()* function of Matlab was used, followed by connecting

endpoints closest to each other to close gaps in the detected outline. After straightening, each image was classified as either as 'egg', or 'worm' using a decision tree-based classifier that was trained on a small subset of manually assigned images. This classification also identified cases where straightening failed (e.g. in the case of self-touching animals), which were removed from further analysis. Details of classifier: The following features were computed for every segmented and straightened image: length, standard deviation of width, cv of width, maximal width, median width, maximal width/median width, volume, volume/length, entropy of width. These features were then used to train an ensemble of 20 bagged (standing for "bootstrap aggregation") decision trees using the *TreeBagger()* function of Matlab. For classification of new data, the same features were calculated, and images were classified using the trained ensemble of decision trees."

- Figure S6 is missing annotation of the larval stages

The missing annotation was added to this figure, which is now called Figure S8.

- It's mainly a matter of taste but, to improve readability (and broaden the readership) I would avoid 'folder mechanism' in the title because it's a bit jargony and it is introduced for the first time by the authors in this manuscript.

We have now changed the title to: "**Coupling of growth rate and developmental tempo reduces body size heterogeneity in *C. elegans***". We believe this title captures our main finding and will indeed be more accessible to a broader readership.

Reviewer #2 (Remarks to the Author):

Towbin and Grosshans presents their results on the regulation of size uniformity during *C. elegans* development. They investigate if there is a working size control which ensures that at each larvae stage *C. elegans* have a more or less fixed size. The growth measurement and analysis have been carefully done, the finding of a new size controlling "folder" mechanism is relevant and in general the manuscript is nicely written. Although there are a few major issues that needs to be resolved before it could be published:

We thank this reviewer for endorsement of the significance of our work and the comments, which have helped us to further improve our manuscript.

The general connection between size control in single cell eukaryotes and the reduced noise in size at developmental stages is interesting and indeed in L2-L4 the growth is more or less equal to the initial size, which is similar to single celled organisms, but in L1 the growth is much larger than the initial size, which already gives a hint that L1 will be differently handled than other stages.

We now display information on absolute volumes and larval stage durations as the main Figure 1e. (previously in Supplemental information).

These actual changes can be seen only from Fig S4, while Fig S3 is explaining how such classical size control measurement can be converted to the % deviation scaled plots of Fig. 3. This Fig S3 is really important to understand key points, it could be even moved to the main text. A missing panel from this is one where volume added is given on the ordinate (like on all other figs). Here a -1 slope would mean a strong sizer (based on figS4 L1 has a weak sizer, which would compensate size difference in 5 cycles).

We have added a new Figure 3 that explains the definitions of sizer, adder, and folder mechanisms. We agree these concepts are very important and this addition should make the manuscript more accessible to a broad readership.

Figure 3. Definition of sizer, adder, and, folder mechanisms
(related to Supplementary Figure S4)

A sizer is defined as a mechanism where the volume at the larval stage exit is independent of the volume at larval stage entry. For an adder, the absolute added volume within a larval stage is independent of the volume at the beginning of the larval stage. For a folder, the volume fold change per larval stage is independent of the volume at larval stage entry. A folder is expected during exponential growth if the exponential growth rate and duration of growth are independent of the current size.

In Supplemental Figure S4, we detail how by normalization to the population average impacts the expected size relations, and how these relations depend on the average volume fold change per larval stage. Importantly, the lack of correlation between the V_1 and $V_2/\Delta V/FC_V$ for sizer/adder/folder mechanisms, respectively, is not affected by this normalization.

This brings the next point: the cell cycle is a periodic process, so small changes in one cycle can be compensated in more than one cycles. In the developmental process discussed here, the oscillations could come from the developmental oscillations, but the number of these is quite limited. It would be great to hear why the compensation should be at each larval stage not only at a single step. The separate control on L1 suggest this as the crucial step, the folder mechanisms at later stages might be able to work only if coupled with a more precise adder or sizer step. I would like to see the calculation in the style of fig S3 to check how the folder model could keep homeostasis.

We are thankful for this comment, which helped devise new analyses to address this point in more detail. We now provide additional simulations (see lines 226 ff., and Figure 4g) comparing size divergence for 4 different scenarios:

- i. Uncoupled growth throughout development (uncompensated folder).
- ii. an adder during L1, followed by an uncompensated folder from L2 to L4.
- iii. a growth rate compensated folder throughout development
- iv. an adder during L1, followed by a compensated folder from L2 to L4

This analysis shows that the adder mechanism in L1, as well as the coupling of growth and development have a significant effect on size homeostasis. We therefore conclude that the combination of an L1 adder and a growth rate compensated folder are required to maintain body volume uniformity.

As suggested, we also plotted the results of these simulations in the form of former Figure S3 (now Figure S4), which we show below. We believe that a box plot of the CVs is better suited to illustrate the conclusions from our simulations to a general readership. We therefore chose this representation in the revised manuscript.

We now also discuss what benefits this combined mechanism of size control may provide (lines 373 ff), and how the smaller number of volume doublings in *C. elegans* compared to unicellular systems impact the required need for precision in size control (lines 367 ff.):

“The coupling of growth and development reduces volume divergence, but unlike adders and sizers does not entirely prevent the accumulation of size differences during development (Fig. 2). Yet, such perfection may not be required in the case of *C. elegans* development, which involves only 4 larval stages and ~ 5.5 volume doublings, compared to hundreds of cell divisions undergone by bacteria. Nevertheless, also in unicellular systems, size control mechanisms involving growth rate regulation have been observed, in addition to size-dependent scaling of the cell cycle duration^{22,50,51}.”

The mathematical model with a combined positive and negative feedback and how its period is driven by a growth rate resembling parameter nicely adds to the results, but it is not really surprising that the frequency of oscillations can be controlled by a parameter which jointly controls the half-life of both species. This parameter can scale the dynamics of the whole system in this way, it is just not fully clear why growth rate should act such way on the developmental oscillator.

We thank the reviewer for this comment, which allowed us to improve our description of the model assumptions (line 339 ff.), and the discussion of its impact (line 396 ff.). The removal rate of A and R is the sum of the degradation rate and the dilution rate, where the dilution rate is equal to the volume growth rate. If the active degradation rate of R is small compared to the growth rate, then the removal rate is determined by the growth rate alone. Thus, in principle, no additional mechanism is required for the coupling of growth and development. We, of course, do not exclude that additional more complex mechanisms do exist and discuss this possibility in the manuscript (line 403 ff.). Nevertheless, our model provides a parsimonious mechanism that explains the observed data.

Extensive published theoretical studies, which we cite in our manuscript, have indeed analysed the dependence of oscillatory frequencies on model parameters including degradation rates in detail. The insight provided by this current manuscript is that, given these models, in a developmental context the dependence of oscillatory frequency on the growth rate can ensure body size uniformity.

Most importantly, the model, inspired new experiments, which we have conducted following a suggestion by reviewer #3: In the model, increasing the degradation rate of R independent of the growth rate accelerates oscillations. We now show this behaviour experimentally in Fig. 7a,b by accelerating the turnover of the oscillatory transcriptionfactor BLMP-1. Accelerating oscillations speeds up development without changing the growth rate and reduces body volume. We believe our model, and further developments thereof, will continue to be of value for the characterization of other oscillator components in future studies.

The changes between growth and no-growth stages were also observed in old experiments on fission yeast cells (Sveiczzer Journal of cell science 1996 or Fantes Journal of cell science 1977). Interestingly a growth rate change was also observed in these cycles as well.

We are thankful for pointing out this parallel to previous work in fission yeast. We have indeed cited Fantes et al. as one of the first papers introducing the concept of the sizer mechanism. We now also discuss the evidence for growth rate related control of size observed in unicellular systems, among others citing Sveiczzer et al. (lines 371 & 372):

“Nevertheless, also in unicellular systems, size control mechanisms involving growth rate regulation have been observed, in addition to size-dependent scaling of the cell cycle duration^{22,50,51}.”

We note that despite similarities, several aspects differ in the size control between yeast and *C. elegans*. Most importantly, unlike Fantes et al., we do not observe an anti-correlation between the added volume and the volume at the beginning of a larval stage. Also, unlike Fantes et al., we find that there is positive auto correlation of the larval stage duration between consecutive stages, whereas Fantes et al. found a (weak) negative auto-correlation of the cell cycle duration between consecutive cell divisions. These differences suggest that distinct mechanisms are at play in unicellular and multi-cellular systems. We believe these differences underline the pertinence of our approach to study size control in the context of an entire animal.

In summary, my main concern is: how far the size control of cell cycle can be correlated with the 4 stages of *C. elegans* developmental program. It was clearly shown that some size control mechanism works, but I am not fully convinced if all 4 stages have a critical role in size homeostasis and if the proposed folder model could indeed serve as a critical size regulator.

As discussed above, we have added additional computational analyses (shown in Fig. 4g) that now quantitatively compare different size control scenarios and the resulting body volume divergence. This analysis shows that the L1 adder, as well as the anti-correlation of growth and the duration of development in L2 to L4 contribute to body volume uniformity. We also provide additional experimental evidence (shown in Fig. 7) that the oscillatory frequency and the duration of development impact the body volume.

Reviewer #3 (Remarks to the Author):

This manuscript by Towbin and Grosshans describes a detailed examination of growth and body size across development in *C. elegans*. The methodology is strong and analyses solid, and the conclusions

made are robust based upon state-of-the-art single animal imaging that is innovative. Much of the manuscript is spent characterizing wild type animals and accommodations for heterogeneity. A general weakness of the manuscript is that it is largely descriptive and only at the very end begins to get into the genetic underpinnings of body volume and growth rate control. I think the work is well conducted but is missing more detailed mechanism and thus based upon its more descriptive nature may be better suited for a specialty journal without the wide readership of Nature Communications.

We thank the reviewer for the endorsement of data quality and the methodological innovation. We believe our study can make an important contribution to the emerging field of organismal systems biology which wants to bridge scales and to apply quantitative approaches established at cellular scale to an organismal context.

To strengthen the molecular insight provided by our study, we have added new experimental data showing that the body size can be predictively altered by increasing the turnover rate of the oscillatory transcription factor BLMP-1. This molecular intervention accelerated gene expression oscillations independent of the growth rate, and thereby reduced body size, consistent with model predictions (Fig. 7, see details in answers below).

In addition to identifying new molecules involved in body size control, we see the importance of our work in providing a quantitative context for future and past molecular and genetic experiments. The quantitative experimental approach and the mathematical modelling framework established by our study, will powerfully support the identification and characterization of further molecular components of the *C. elegans* oscillatory clock in future research.

Finally, we believe that the novel and fundamental insight that a developmental oscillatory clock can control body volume and that intrinsic properties of a genetic oscillator can ensure body volume uniformity will be of great interest to a broad and interdisciplinary readership.

The analyses are done uniformly at one temperature, and one at which many believe *elegans* is under heat stress. The authors should give thought to the effect of temperature on the process and whether the same relationships hold true.

In Figure S3 and S6, we now show additional experiments showing that trends observed at 25 °C also hold true at 20 °C.

i. The heterogeneity of growth and size at different larval stages (Fig. S3)

Like at 25°C, the volume divergence among individuals during development is small at 20°C, even a bit smaller than at 25°C. This reduced heterogeneity among individuals is consistent with the notion that 25°C is close to the upper bound of the temperature range compatible with growth of *C. elegans*. We now discuss the temperature dependence of growth and size heterogeneities in the main text (line 176 ff.) and show the new experimental data in Figure S3:

“Volume divergence among individuals at 20°C was even smaller than at 25°C, consistent with 25°C being close to the upper bound of the temperature range compatible with growth of *C. elegans* (Fig. S3).”

Figure S3. Volume divergence among individuals and growth rate heterogeneity at 20°C

(top) Histogram of the growth rate deviations in % from the population mean measured at indicated larval stages at 20°C. The growth rate of each individual and larval stage was determined by a linear regression to $\log(\text{volume})$ against time excluding the first 10% and the last 25% of the larval stage.

(bottom) Histogram of body volume deviations in % from the mean measured at birth and indicated larval moults at 20°C. Coefficients of variation are indicated above each panel.

ii. Distinction of sizer, adder, and folder mechanisms at 20 °C (Fig. S6)

Figure S6 shows analysis of sizer, adder, and folder models at 20 °C. We find that overall trends are robust to changes in temperature. We do observe small quantitative differences between measurements conducted at 25°C and at 20°C. Specifically, there is a slightly more negative slope between the fold change and the size at the start of the larval stage in L2 and L4 animal at 20 °C (-0.38 and -0.35 at 20°C vs. -0.27 and -0.13 at 25°C), indicating a stronger size dependent contribution to growth control at L2 and L4 at lower temperature. We now discuss the contribution of this weak size dependence in addition to the coupling of growth and development in the text (line 201 ff.):

“These trends were robust to changes in temperature: like at 25 °C, V_2 and ΔV were also positively correlated with V_1 at 20 °C (Fig. S6). We observed a slightly stronger negative relation between V_1 and FC_V for L2 and L4 stages (-0.69, -0.38, 0.03, -0.35 at 20 °C vs. -0.61, -0.27, -0.14, -0.13 at 25 °C for L1 to L4, Fig. S6), indicating a weak size dependence of volume growth at lower temperature.”

Figure S6. adder, sizer, folder models at 20 °C

(related to Figure 4)

Scatter plot of volume at larval stage entry vs. volume at larval stage exit shown as % deviation from the mean for indicated larval stages. Colour indicates point density. Red circles are a moving average along x-axis. Thick red trendline: robust linear regression to the data (see methods). Thin red lines: 95% CI. Slope +/- 95% CI of the trendline is indicated above the panels.

The heterogeneity among the nutritional model used, *eat-2* mutants, is far greater than the other genetic mutants. Whether this is an intrinsic property of this mutant or of the effects of nutrient deprivation on the “oscillator” as the authors propose for wild type animals, could be tested by using diets of different “quality” e.g. as in MacNeil et al Cell 2013.

We fully agree that the increased heterogeneity in growth rate of the *eat-2* mutant could be a general effect of dietary restriction, or an intrinsic property of this specific mutant. We did not mean to make any statement regarding the origin of this increased heterogeneity in our initial submission and believe we have also not done so. To improve the clarity in this regards, we now explicitly state these two possibilities in the text (line 288):

“We cannot distinguish if the increased heterogeneity among *eat-2* mutant individuals is a general effect of dietary restriction, or a characteristic of this specific mutant.”

Nevertheless, as suggested, we have conducted experiments using different bacterial diets, focusing on L2 and L3 stages for technical reasons. Consistent with previous experiments, we find that the growth rate depends on the bacterial strain used as a diet (Avery and Shtonda, 2003; McNeil et al., 2013). Consistent with previous studies (Avery and Shtonda, 2003) the growth effects of the *eat-2* mutation is much stronger than the effects of bacterial food sources, although previous work only measured developmental rates (defined as $1/[\text{developmental time}]$) and not growth rates ($d\log[\text{Volume}]/dt$), prohibiting a direct quantitative comparison.

Worms growing on the food promoting the slowest growth (NA22) also have a slightly higher growth rate heterogeneity than worms growing on OP50. However, since the range of growth rates that can be explored by different bacterial strains is small, and the molecular cause of the growth rate heterogeneity of the *eat-2* mutant is not the focus of our current manuscript, we prefer not to include this additional data in our revised manuscript.

We believe that our explicit statement that phenotypes of the *eat-2* do not necessarily generalize to other forms of dietary restriction (see above) is an important addition to the text and will prevent any misunderstandings by our readers.

	CV growth rate			n	
	L2	L3		L2	L3
OP50	6.67%	7.66%		91	56
HB101	6.87%	6.55%		61	57
DA837	9.39%	5.21%		60	35
DA1877	7.91%	8.85%		98	87
NA22	7.95%	8.74%		85	52
OP50 eat-2	17.49%	22.48%		295	297

References:

Avery, L. & Shtonda, B. B. Food transport in the *C. elegans* pharynx. *The Journal of experimental biology* **206**, 2441–2457 (2003).

MacNeil, L. T., Watson, E., Arda, H. E., Zhu, L. J. & Walhout, A. J. M. Diet-Induced Developmental Acceleration Independent of TOR and Insulin in *C. elegans*. *Cell* **153**, 240–252.

The testing of a heterochronic mutant is a nice touch, but in general the loss of machinery responsible for cyclic gene expression (oscillatory) is not explored, realizing the limitation that we do not fully

understand the machinery necessary for oscillations in the worm. None the less, circadian gene expression has been described in detail in the worm (Goya et al. PNAS 2016; Benerjee et al Dev Cell 2005; review in Hasegawa et al Chronobiol Int. 2005), thus the possibility of entrainable oscillations and rhythms that the authors could explore mechanistically in more detail.

We thank the reviewer for this suggestion, which helped us to design new experiments now shown in Figure 7 that address this point. Before we explain these results, we would like to respectfully point out a potential misconception: the developmental oscillator of *C. elegans* is not a circadian clock. Its period is ultradian (e.g., 7-8 h at 25 °C) rather than circadian (~24 h), and not temperature-compensated (it increases as ambient temperature decreases; Hendriks et al., Mol Cell 2014, Kim et al., Nat Genet 2013). Entrainment has not been reported either (Goya et al. studied circadian, not developmental rhythms). Finally, the heterochronic pathway studied by Banerjee et al. is a 'linear' timer that controls stage-specific changes in cell fates; how this connects mechanistically to the oscillator is unclear. Hence, explorations of the circadian machinery and entrainable rhythms do not offer a way forward in molecular dissection of the developmental clock that we studied here. However, the reviewer's comment motivated us to examine the effect of perturbing oscillations through modulation of BLMP-1, currently the only validated clock component (Hauser et al., bioRxiv 2021).

We now show that alteration of the oscillatory frequency by increasing the degradation rate of the oscillatory transcription factor BLMP-1 accelerates development without changing the growth rate and thereby reduces body size (Fig. 7a,b). This experiment validates a molecular link between gene expression oscillations, growth rate, and body size.

Figure 7. Changing the oscillatory frequency modulates body volume

a. *dpy-9p::gfp* expression of strain with *blmp-1::aid*, *eft-3p::tir1* at indicated auxin concentrations. Averaging of individuals was conducted as described in Fig. 6a. GFP is shown as intensity per pixel. circle, square, triangle, and cross indicate M1-M4 in the respective condition.

b. Same as (a), but for volume.

c. Mathematical model of genetic oscillator based on design II of ref. ⁴⁴. Activator *A* activates its own production and the production of the repressor *R* by a factor of ρ . *R* degrades *A* at a rate σ . β and γ are the basal production rates of *A* and *R*, respectively. μ is the growth rate. *R* is considered stable, such that its removal rate is set by the growth rate μ . Model parameters ($\beta = 10/h$, $\gamma = 0.3/h$, $\sigma = 10/h$, $\rho = 50$) are close to a saddle-node on invariant circle (SNIC) bifurcation ⁴⁵ as is experimentally observed ²³. Modelled dynamics of *A* (dotted lines) and *R* (solid lines) for $\mu = 0.12/h$ (blue), and for $\mu = 0.10/h$ (red).

d. Impact of growth rate on the removal rate of a protein. In a rapidly growing individual (blue, left), a stable protein dilutes more quickly than in a slowly growing individual. The dilution rate is equivalent to the growth rate μ in the model (c).

e. Oscillation frequency as a function of the growth rate (red). Missing values at slow and fast growth rates are due to bifurcation to a non-oscillating state. Black dotted line: proportionality between the growth rate and the oscillation frequency, which would ensure perfect size homeostasis. Black circle: reference growth rate at which proportional scaling is precise for the given parameters ($\mu = 0.11/h$). Insert shows the same data but mean normalized to the reference growth rate $\pm 20\%$, equivalent to the display of experimental observations shown in Figure 4 & 6.

f. Same as (e), but for growth rate vs. volume fold change. The invariance of the volume fold change with respect to the growth rate is qualitatively independent on parameters over a wide range (see Fig. S10).

References:

Hendriks, G. J., Gaidatzis, D., Aeschimann, F. & Grosshans, H. Extensive oscillatory gene expression during *C. elegans* larval development. *Molecular cell* **53**, 380–92 (2014).

Kim, D. H., Grün, D. & van Oudenaarden, A. Dampening of expression oscillations by synchronous regulation of a microRNA and its target. *Nature genetics* **45**, 1337–44 (2013).

Goya, M. E., Romanowski, A., Caldart, C. S., Bénard, C. Y. & Golombek, D. A. Circadian rhythms identified in *Caenorhabditis elegans* by in vivo long-term monitoring of a bioluminescent reporter. *PNAS* **113**, E7837–E7845 (2016).

Banerjee, D., Kwok, A., Lin, S.-Y. & Slack, F. J. Developmental timing in *C. elegans* is regulated by kin-20 and tim-1, homologs of core circadian clock genes. *Dev Cell* **8**, 287–295 (2005).

Hauser, Y. P., Meeuse, M. W. M., Gaidatzis, D. & Großhans, H. *The BLMP-1 transcription factor promotes oscillatory gene expression to achieve timely molting.* <http://biorxiv.org/lookup/doi/10.1101/2021.07.05.450828> (2021) doi:10.1101/2021.07.05.450828.

- The major question as to whether the oscillations are governed by growth or whether the oscillations govern growth is a critical one to answer. Presumably there will be oscillatory genes/proteins in both classes.

We address this major question by new data shown in Figure 7. Using quantitative modulation of BLMP-1 turnover by inserting an auxin inducible degradation (AID) tag, we show that the frequency of *dpy-9p::gfp* oscillations can be altered independent of changes to the growth rate, suggesting that the oscillations do not govern growth directly. However, the larval stage duration was shortened upon acceleration of oscillations, indicating that oscillations do govern the rate of development.

We agree that this finding opens opportunities for many exciting new projects and future studies, including the investigation of possibly distinct responses of individual genes or tissues to accelerated development.

It may be possible to use genes/proteins that are expressed uniquely at the molt (e.g. *mlt-10*, *mlt-12*) to test the specific features of the folder hypothesis that the authors put forward. By their rationale, the expression of these proteins should only appear when the criteria specified by a folder mechanism are met.

We thank the reviewer for pointing out our lack of clarity that prompted a misunderstanding: We do indeed not propose that the expression of moulting genes, and thus the moult itself, is triggered by a mechanism that detects a target fold change. Instead, we propose that the coupling of growth and development is a continuous process that is intrinsic to the growth dependent dilution of the oscillator

components, as described by our mathematical model in Fig. 7c and now graphically illustrated in a new Fig. 7d. We have revised the discussion section to explain this point more clearly (line 392 ff.):

“According to this latter model, the coupling of growth and development is a continuous process that occurs throughout the larval stage, rather than a singular event occurring at the larval moult. Such continuous coupling is also consistent with proportional scaling of developmental events among individuals⁵² and the temporal scaling of gene expression oscillations at reduced temperature²⁴. “

REVIEWERS' COMMENTS

Reviewer #2 (Remarks to the Author):

The authors have addressed my concerns and the updated MS explains the main claims in a much more convincing way. I appreciate the addition of new figures (especially Fig 7) responding to comments of other referees as well. The MS is now acceptable for publication.

Reviewer #3 (Remarks to the Author):

The authors have addressed the entirety of my concerns, experimentally when possible. I am now in favor of publication of this high quality and interesting manuscript in Nature Communications. I have only a few small suggestions on verbiage:

Line 178 the authors may want to say "compatible with robust control of body size of *C. elegans*." Rather than just "growth".

The BLMP-1 data are compelling with the following caveat: the supposition by the authors that low dose Auxin is capable of dose-titrating blmp-1 levels is not justified by recent observations that even low doses of Auxin (down to 10 micromolar) are quite effective in promoting protein degradation, e.g. <https://www.biorxiv.org/content/10.1101/2021.08.12.456148v1>
It would solidify the argument significantly to see protein levels associated with these auxin doses.

Point-by-point response to reviewer's comments

Reviewer #2 (Remarks to the Author):

The authors have addressed my concerns and the updated MS explains the main claims in a much more convincing way. I appreciate the addition of new figures (especially Fig 7) responding to comments of other referees as well. The MS is now acceptable for publication.

We thank this reviewer for the positive evaluation of our revision and for the comments that helped us improve our manuscript.

Reviewer #3 (Remarks to the Author):

The authors have addressed the entirety of my concerns, experimentally when possible. I am now in favor of publication of this high quality and interesting manuscript in Nature Communications. I have only a few small suggestions on verbiage:

We thank this reviewer for the positive evaluation of our revision and for the comments that helped us improve our manuscript.

Line 178 the authors may want to say "compatible with robust control of body size of *C. elegans*." Rather than just "growth".

Done.

The BLMP-1 data are compelling with the following caveat: the supposition by the authors that low dose Auxin is capable of dose-titrating blmp-1 levels is not justified by recent observations that even low doses of Auxin (down to 10 micromolar) are quite effective in promoting protein degradation, e.g. <https://www.biorxiv.org/content/10.1101/2021.08.12.456148v1>. It would solidify the argument significantly to see protein levels associated with these auxin doses.

We thank this reviewer for this comment and agree that it would be informative to measure the titration of BLMP-1 concentrations directly, rather than its phenotypic consequences. Unfortunately, such an experiment is technically not achievable at sufficient accuracy under the experimental conditions used here (inside single animal chambers). The strain used in this study does not have a fluorescent tag, and green and red fluorophores are indeed already used for other purposes in this experiment. Indeed, we know from looking at a previously published endogenously GFP-tagged BLMP-1 (Stec et al., Curr Biol 2021), that the signal even in an unperturbed condition is barely above background, further hindering quantification.

The effectiveness of IAA in protein degradation is expected to depend strongly on the specific protein targeted (e.g. due to its baseline turnover rate or the number of lysins in favourable geometry), such that extrapolations from other proteins are not necessarily informative. We note however, that also in the cited paper by Smith et al, there is a clear dosage effect of IAA detectable in Fig S2A and mentioned by the authors, especially in younger animals. Indeed, in our own experience, we have found many examples where titration of IAA has a gradual molecular and

phenotypic effect, and where the relevant dose that induces a phenotypic effect seems to be very protein specific (ranging from sub-micro molar to >1mM for reaching full depletion).

We now explicitly state in the manuscript, that it was not possible to measure BLMP-1 levels directly on ln. 324: “We could thereby titrate the degradation rate of BLMP-1 using a high (1mM) and at a low (16 μ M) dose of auxin in micro chambers, although due to the low basal expression level of BLMP-1 it was not possible to measure BLMP-1 levels directly.”